# Unprecedented Fe delivery from the Congo River margin to the South Atlantic Gyre

Lúcia H. Vieira [1]*, Stephan Krisch [1], Mark J. Hopwood [1], Aaron J. Beck [1], Jan Scholten[2], Volker Liebetrau[1] & Eric P. Achterberg [1]

Rivers are a major supplier of particulate and dissolved material to the ocean, but their role as sources of bio-essential dissolved iron (dFe) is thought to be limited due to rapid, efficient Fe removal during estuarine mixing. Here, we use trace element and radium isotope data to show that the influence of the Congo River margin on surface Fe concentrations is evident over 1000 km from the Congo outflow. Due to an unusual combination of high Fe input into the Congo-shelf-zone and rapid lateral transport, the Congo plume constitutes an exceptionally large offshore dFe flux of $6.8 \pm 2.3 \times 10^8$ mol year$^{-1}$. This corresponds to $40 \pm 15\%$ of atmospheric dFe input into the South Atlantic Ocean and makes a higher contribution to offshore Fe availability than any other river globally. The Congo River therefore contributes significantly to relieving Fe limitation of phytoplankton growth across much of the South Atlantic.

---

[1] GEOMAR Helmholtz Centre for Ocean Research Kiel, Wischhofstr. 1-3, 24148 Kiel, Germany. [2] Institute of Geosciences, University of Kiel, Otto-Hahn-Platz 1, 24118 Kiel, Germany. *email: lvieira@geomar.de

Elevated dissolved trace element (dTE; defined by <0.2 μm filtration) concentrations in coastal regions are derived from riverine inputs[1,2], benthic pore-water and resuspended sediment supply[3], atmospheric deposition[4], and submarine groundwater discharge (SGD)[5]. Whilst riverine fluxes of dTEs into the ocean are significant, estuarine processes remove a high, but variable, fraction of riverine dissolved Fe (dFe). Typically 90–99% of dFe is removed at low salinity in estuaries due to the rapid aggregation of Fe and organic species with increasing ionic strength[6], with further removal by biological uptake and scavenging in estuarine and shelf regions[7]. Whilst riverine Fe concentrations are 3–5 orders of magnitude greater than those in seawater[8], rivers provide only ~3% (after estuarine removal) of the new Fe delivered annually to the oceans[9]. Consequently, there is typically limited potential for river-derived Fe to directly affect productivity in offshore ocean regions where Fe often limits, or co-limits, primary production[10].

The Congo is the second largest river on Earth by discharge volume[11], and is the only major river to discharge into an eastern boundary ocean region with a narrow shelf[12]—unique characteristics for a near-equatorial river plume subject to low Coriolis forces[11]. Although the Congo is a large source of freshwater to the SE Atlantic[13], little is known about associated TE fluxes and their influence on SE Atlantic productivity. In November–December 2015, the GEOTRACES cruise GA08 proceeded along the SW African shelf to determine the lateral extent of chemical enrichment from the Congo plume. Here we use a conservative terrigenous tracer, naturally occurring radium isotopes ($^{228}$Ra and $^{224}$Ra), in combination with TE distributions to derive TE fluxes from the Congo plume into the South Atlantic. Radium isotopes are produced by sedimentary thorium decay, released from river plumes and shelf sediments, and then transported to the open ocean by turbulent mixing and advection[14,16]. In seawater, only mixing and decay processes control Ra distribution.

Here we show that the dFe removal occurring within the Congo estuary is balanced by other dFe inputs into the shelf region near the Congo River outflow (Congo-shelf-zone; Fig. 1). River input of dissolved and desorbed particulate Ra cannot maintain the Ra inventory observed in the Congo-shelf-zone. There must be a significant Ra and TE source between the river and the Congo-shelf-zone, likely shelf sediments and/or SGD. These inputs, combined with strong lateral advection, sustain a pronounced plume of dTEs evident in an off-shelf transect at 3°S. The presence of TEs and $^{228}$Ra, and their strong inverse correlation with salinity in surface waters several hundreds of kilometers off-shelf indicates rapid horizontal mixing of the river plume. This facilitates the delivery of high levels of TEs from the Congo River outflow into the Southeast Atlantic Ocean, a reportedly oligotrophic region[10].

## Results and Discussion

**Trace element and radium distributions**. On the shelf where Congo waters first encounter the Atlantic Ocean, hereafter the Congo-shelf-zone (Fig. 1), the mean dFe concentration was ~15% of the Congo River concentration, indicating low apparent dFe removal compared with Congo River freshwater. About 50–85% of river-derived dFe is reportedly removed from solution at low salinities (0–5) in the Congo estuary, with the greatest removal in large size fractions[2]. Mean (± standard deviation) dFe concentration in the Congo River freshwater (measured in April, July, and October 2017) was 7380 ± 3150 nM, similar to limited previous measurements (~9000 nM)[2]. Extrapolating the linear regression line of dFe vs. salinity in the Congo-shelf-zone (Fig. 2a) to zero salinity provides an effective-zero-salinity-endmember concentration of 3910 ± 610 nM ($R^2 = 0.76$), indicating that only ~50% of dFe is removed during estuarine mixing processes. This is consistent with prior work[2], but notably limited compared with other river systems where 90–99% is typically stripped from the water column[2,6]. Slow removal of Fe in some

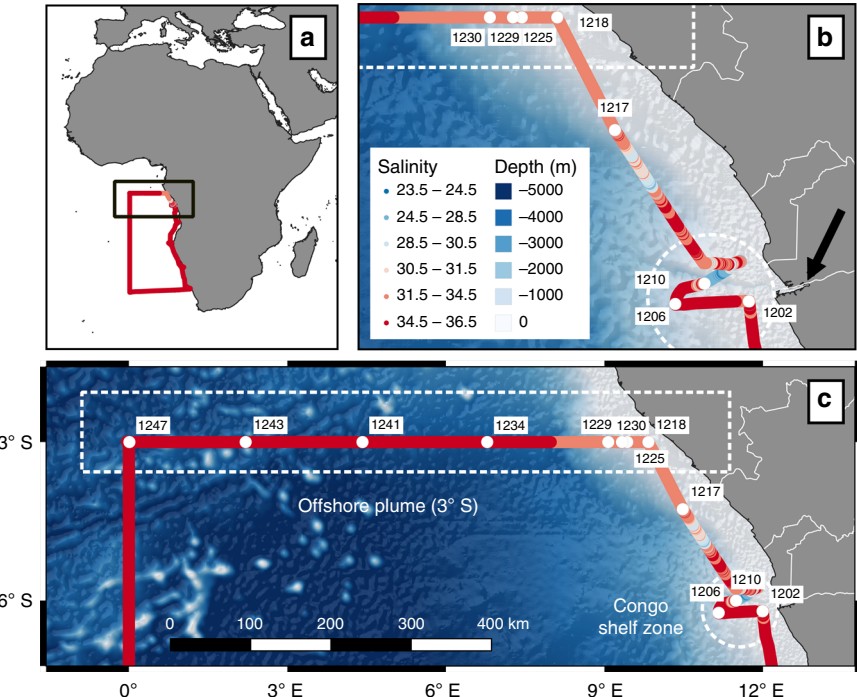

**Fig. 1 Map of the study region showing station locations along the GEOTRACES GA08 cruise transect.** Ship underway salinity measurements along the cruise section (**a**). Insets provide a detailed view of the study region including bathymetry and $^{228}$Ra stations sampled in the Congo-shelf-zone (**b**; stations 1202, 1206, and 1210), along the coastal transect (between stations 1210 and 1218), and along the offshore 3°S transect (**c**; stations 1218–1247). Satellite-derived surface seawater salinity during GEOTRACES cruise GA08 is provided in the Supplementary Information. Bathymetry and shoreline data were obtained from ref. [65] and ref. [66].

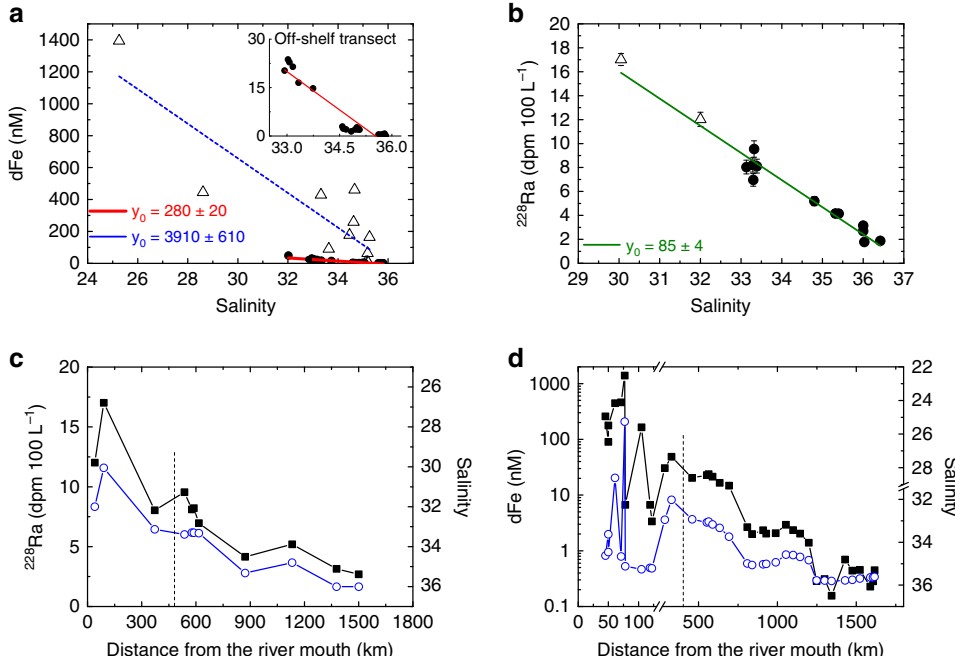

**Fig. 2 Distributions of dFe, Ra and salinity in surface waters.** Mixing diagram between river and open ocean waters from the Congo-shelf-zone to the end of the 3°S transect (stations 1202-1247) for dFe (**a**) and $^{228}$Ra concentrations (**b**). Open triangles represent the samples collected in the Congo-shelf-zone and circles represent the samples in the off-shelf 3°S transect. Dashed blue line in **a** represents the regression line for the Congo-shelf-zone, and the red line represents linear regression for the off-shelf transect. Intercepts are represented as $y_0$ and considered as the effective-zero-salinity-endmembers. Inset in **a** is an expanded version of the off-shelf transect. The regression line for $^{228}$Ra (green line in **b**) includes all data. **c**, **d** $^{228}$Ra, dFe (note log scale) (solid squares), and inverse salinity distributions (open circles) in plume surface waters from the Congo-shelf-zone to the end of the 3°S transect (stations 1202-1247). Dashed vertical lines in **c** and **d** represent the beginning of the off-shelf transect at 3°S. Source data are provided as a Source Data file.

**Table 1 Radium-228 and trace element fluxes.**

| | Congo River[a] | Congo-shelf-zone[b] | Off-shelf 3°S transect[b] |
|---|---|---|---|
| $^{228}$Ra Flux (atoms year$^{-1}$) | $4.8 \pm 0.4 \times 10^{21}$ | $3.4 \pm 0.9 \times 10^{21}$ | $6.2 \pm 2.0 \times 10^{21}$ |
| dFe-Flux (mol year$^{-1}$) | $9.6 \pm 4.1 \times 10^{9}$ | $5.6 \pm 4.6 \times 10^{9}$ | $6.8 \pm 2.3 \times 10^{8}$ |
| dMn-Flux (mol year$^{-1}$) | $1.3 \pm 0.4 \times 10^{8}$ | $4.4 \pm 1.8 \times 10^{8}$ | $3.1 \pm 1.2 \times 10^{8}$ |
| dCo-Flux (mol year$^{-1}$) | $2.2 \pm 0.8 \times 10^{6}$ | $5.3 \pm 2.0 \times 10^{6}$ | $4.6 \pm 1.8 \times 10^{6}$ |
| dFe/$^{228}$Ra gradient (pmol atoms$^{-1}$) | – | $1.66 \pm 1.3$ | $0.1 \pm 0.02$ |
| dMn/$^{228}$Ra gradient (pmol atoms$^{-1}$) | – | $0.13 \pm 0.04$ | $0.05 \pm 0.01$ |
| dCo/$^{228}$Ra gradient (fmol atoms$^{-1}$) | – | $1.6 \pm 0.5$ | $0.7 \pm 0.2$ |
| Average $^{228}$Ra activity (dpm 100 L$^{-1}$) | – | $12.7 \pm 3.6$ | $8.62 \pm 0.86$ |
| Average dFe concentration (nM) | $7380 \pm 3150$ | $920 \pm 670$ | $20.9 \pm 1.67$ |
| Average dMn concentration (nM) | $105 \pm 30$ | $73 \pm 4.0$ | $14.7 \pm 4.48$ |
| Average dCo concentration (nM) | $1.72 \pm 0.59$ | $0.88 \pm 0.05$ | $0.20 \pm 0.03$ |

[a]Trace element (TE) fluxes from the Congo River were determined using the measured TE concentrations in the Congo River (Supplementary Table 1) and river discharge reported in ref. [30]. Radium-228 flux from the Congo River was estimated by extrapolating the regression line to zero salinity (Fig. 2b) and multiplying the intercept by the river discharge
[b]See "Methods" for details on TE and $^{228}$Ra concentrations and fluxes in the Congo-shelf-zone and off-shelf transect

estuaries[17] has been attributed to stabilization by organic ligands, making the dFe pool resistant to flocculation[18]. Alternatively, sources of Fe other than river water may simply offset the loss from estuarine mixing. Indeed, similar $^{228}$Ra and other TE enrichments over the Congo-shelf-zone (Fig. 2b; Supplementary Fig. 1) suggest they have a common source, likely shelf sediments[3,19] or SGD[5]. This indicates that the apparently low removal primarily reflects additional sources of dFe rather than unusually pronounced stabilization of the river-derived dFe. Strong benthic Fe input in this region is consistent with observed high rates of sediment accumulation and high rates of iron solubilization from the shelf break down to deep-sea fan sediments[20,21], which may also contribute to the relatively high dFe concentrations in the Congo-shelf-zone.

Covariations of dissolved manganese (dMn) and cobalt (dCo) with salinity indicate an effective-zero-salinity-endmember higher than dMn and dCo concentrations measured in the river (Table 1). This is also indicative of nonconservative dTE inputs into the Congo-shelf-zone relative to simple mixing of river and seawater (Supplementary Fig. 1). An additional TE source in the Congo-shelf-zone is also evident in the lower Fe/Mn ($6.3 \pm 6.0$) and Fe/Co ($525 \pm 490$) ratios compared with the Congo River (Fe/Mn $= 71.2 \pm 37.5$; Fe/Co $= 4.29 \pm 2.34$). These ratios reflect how dFe is removed relative to the other elements, which is unclear from the available gross fluxes alone. These ratios suggest that Congo River dFe is removed by a factor of 10, whereas the fluxes (Table 1, discussed below) indicate that the removal is only a factor of 2. In summary, a multielement approach also

corroborates significant dTE inputs into the Congo-shelf-zone other than Congo River water.

Elevated $^{228}$Ra and TEs in the Congo River plume can be traced off-shelf at 3°S over 1000 km from the Congo River mouth (Fig. 2c). Benthic input supplies $^{228}$Ra and TEs between the river mouth and the Congo-shelf-zone (here considered as the 10,000 km$^2$ plume area; "Methods"), but conservative $^{228}$Ra mixing behavior (Fig. 2b) then indicates no additional $^{228}$Ra inputs beyond the Congo-shelf-zone. The similarity between the $^{228}$Ra and TE distributions with salinity (Fig. 2c, d; Supplementary Fig. 1) indicates that the plume forms the only major source of Ra and TEs in this region. Salinity increased offshore along the 3°S transect, with occasional fresher pockets coincident with elevated $^{228}$Ra and TE concentrations (e.g., at 1100 km; Fig. 2c, d). These likely result from transport by filaments, meanders, or eddies originating near the Congo River mouth[22,23]. The linear $^{228}$Ra gradient with distance beyond 360 km offshore ($R^2 = 0.91$) (Supplementary Fig. 2) indicates that the $^{228}$Ra distribution is controlled by eddy diffusion near the shelf break[24] (Supplementary Note 1). The slope of $^{228}$Ra vs. distance changes beyond the shelf break, likely due to offshore advection.

The effective-zero-salinity-endmember concentration calculated for dFe was 90% lower for the off-shelf samples compared with the Congo-shelf-zone samples (280 nM vs. 3910 nM; Fig. 2a), indicating substantial removal of dFe along the flow path of the Congo plume within 400 km of the river mouth. Despite this removal, elevated dFe was still observed at the start of the off-shelf transect, up to 600 km from the river mouth (~25 nM; Fig. 3). dFe concentrations in other shelf systems typically decline sharply at the shelf break to less than 1 nM[25,26] and thus the influence of major rivers (e.g., the Amazon River) on surface ocean Fe concentrations is more limited[27]. dFe concentrations of ~15 nM were observed 100 km beyond the shelf break, and remained >2 nM for 500 km beyond the shelf break. These features indicate a sustained Fe flux into the South Atlantic Gyre where Fe limitation or co-limitation of primary production has been observed[10].

**Radium and trace element fluxes in the Congo-shelf-zone.** Ra isotopes were used to quantify TE fluxes for the Congo-shelf-zone, where additional TE and Ra inputs created an intermediate endmember that mixed approximately conservatively along the Congo plume (Fig. 2a, b); and also for the 3°S off-shelf transect, where there was no additional Ra input.

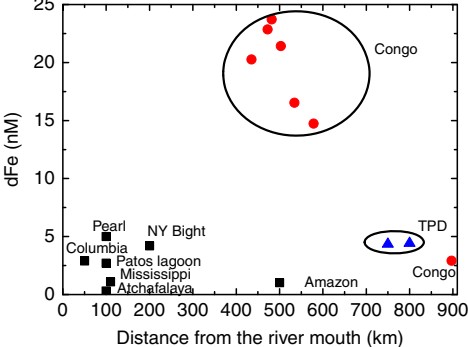

**Fig. 3 Comparison of dFe concentrations vs. distance from the river mouth in other riverine systems globally.** The presence of dFe several hundreds of kilometer off-shelf indicates more rapid horizontal mixing of Congo River plume compared with other systems[1,5,27,54,67–69]. TPD indicates transpolar drift, and NY Bight indicates New York Bight. Source data are provided as a Source Data file.

The highest $^{228}$Ra concentration ($^{224}$Ra = 8.15 dpm 100 L$^{-1}$; $^{228}$Ra = 17.2 dpm 100 L$^{-1}$) was found at the lowest observed salinity ($S = 30$; 100 km from river mouth; Fig. 2b). At salinity 30, all surface-associated Ra from river particles is desorbed[16,28,29] (Supplementary Fig. 3). The $^{228}$Ra Congo-shelf-endmember ("Methods") therefore includes all dissolved $^{228}$Ra derived from desorption from river-borne particles, the river dissolved phase, and shelf sediments near the river mouth. The water residence time in the Congo-shelf-zone is ~3 days[13,30]. Assuming steady state and negligible loss by decay, the residence time and $^{228}$Ra inventory in the Congo-shelf-zone ("Methods"), indicate a $^{228}$Ra flux into this region of $3.4 \pm 0.9 \times 10^{11}$ atoms m$^{-2}$ year$^{-1}$, or ~$3.4 \pm 0.9 \times 10^{21}$ atoms year$^{-1}$ when scaled to a plume area of 10,000 km$^2$.

Conservative mixing between the Congo-shelf-endmember and offshore waters (Fig. 2b) indicates a riverine $^{228}$Ra effective-zero-salinity-endmember concentration of $85 \pm 4$ dpm 100 L$^{-1}$, including both dissolved and desorbed Ra. Together with the river discharge ($1.3 \times 10^{12}$ m$^3$ year$^{-1}$)[31], this suggests a fluvial $^{228}$Ra flux of $4.8 \pm 0.4 \times 10^{21}$ atoms year$^{-1}$, which is similar to the $^{228}$Ra flux estimated for the Congo-shelf-zone ($3.4 \pm 0.9 \times 10^{21}$ atoms year$^{-1}$). If the assumption of conservative mixing behavior is correct, the effective-zero-salinity-endmember would be similar to actual river Ra concentrations, which are not known.

Radium-228 data are not available for the Congo River, but global rivers are generally less than 20 dpm 100 L$^{-1}$[32] (Supplementary Fig. 3). Assuming the Congo is similar to other major rivers (~20 dpm 100 L$^{-1}$), the average Congo River discharge[31] would supply $2.6 \times 10^{14}$ dpm year$^{-1}$, or ~$10 \times 10^{20}$ atoms year$^{-1}$ of dissolved $^{228}$Ra. Desorption of surface-bound Ra from river-borne particles generally supplies <2 dpm g$^{-1}$[15]. The Congo suspended sediment load is 43 Mt year$^{-1}$[31], so desorption can supply no more than ~$3.75 \times 10^{20}$ atoms year$^{-1}$, or an equivalent dissolved $^{228}$Ra activity of 6 dpm 100 L$^{-1}$. Thus, the total supply of $^{228}$Ra from the Congo River itself is estimated to be ~$1.4 \times 10^{21}$ atoms year$^{-1}$, or only ~30–35% of the determined flux into the Congo-shelf-zone. If the remainder was supplied by benthic diffusion, it would represent a flux on the order of $400 \times 10^9$ atoms m$^{-2}$ year$^{-1}$, nearly fourfold higher than the maximum reported globally[33]. This suggests that either the dissolved $^{228}$Ra concentration in the Congo River is exceptionally high compared with other large rivers (~79 dpm 100 L$^{-1}$ vs. <20 dpm 100 L$^{-1}$ elsewhere); or $^{228}$Ra diffusion from shelf sediments in this region is anomalously high compared with other regions globally; or there is another source of Ra such as SGD[34,35]. Based on observations elsewhere of large variability of $^{228}$Ra diffusion from shelf sediments[36] and SGD input[14], the second and third hypothesis, or a combination of both, are most likely.

Radium-228 and TEs have a common source in the estuarine mixing zone up to our Congo-shelf-endmember, and combining the $^{228}$Ra flux and concentration ratios of TE/$^{228}$Ra ("Methods") for the Congo-shelf-zone provides a dFe-flux (dFe-Flux$_{Congo-shelf}$) of $5.6 \pm 4.6 \times 10^5$ μmol m$^{-2}$ year$^{-1}$ ($5.6 \pm 4.6 \times 10^9$ mol year$^{-1}$), an order of magnitude higher than reported along other continental margins[37,38]. The corresponding dMn-Flux$_{Congo-shelf}$ was $4.4 \pm 1.8 \times 10^8$ mol year$^{-1}$; and dCo-Flux$_{Congo-shelf}$ was $5.3 \pm 2.0 \times 10^6$ mol year$^{-1}$ (Table 1). The large $^{228}$Ra and TE fluxes determined here are derived from their inventories in the Congo-shelf-zone, and this does not take into account seasonal variations in their supply to surface waters. Sample collection at sea occurred during the high discharge season of the Congo River. Seasonal variations of the river flow (mean annual range between 35 and $60 \times 10^3$ m$^3$ s$^{-1}$)[39] with approximately a twofold seasonal variation in dFe concentrations (Supplementary Table 1) can strongly affect the Congo River plume dispersion[40], and potentially the delivery

of river-derived materials to the SE Atlantic Ocean. The corresponding seasonal variation in benthic supply of $^{228}$Ra and TEs to overlying plume waters on the Congo shelf is also unconstrained.

**Radium and trace element off-shelf fluxes at 3°S.** The shelf width at 3°S is 70 km, giving an offshore $^{228}$Ra inventory within the plume cross section of $2.6 \pm 0.2 \times 10^{13}$ atoms m$^{-2}$. With a residence time of $7 \pm 2$ days ("Methods"), a $^{228}$Ra input from the Congo plume into the open Atlantic Ocean is $1.4 \pm 0.4 \times 10^{15}$ atoms m$^{-2}$ year$^{-1}$, or assuming a plume thickness of 15 m (Supplementary Fig. 4), $22 \pm 6.2 \times 10^{15}$ atom year$^{-1}$ m-shoreline$^{-1}$. Satellite-derived surface salinity at 3°S (Supplementary Fig. 5) indicates a shoreline plume width of ~300 km, so the total offshore $^{228}$Ra flux is $6.2 \pm 2.0 \times 10^{21}$ atoms year$^{-1}$. Within uncertainties, this flux matches that estimated for the Congo-shelf-zone ($3.4 \pm 0.9 \times 10^{21}$ atoms year$^{-1}$), consistent with the observed conservative mixing behavior of $^{228}$Ra in the plume. The flux represents 4% of previous estimates of total annual $^{228}$Ra input into the Atlantic Ocean[33].

Tracer derived flux calculations in dynamic cross-shelf regions are complicated by meso-scale features like eddies[23], which may influence TE and Ra distributions. Nevertheless, our data unambiguously indicate that the Congo plume is a dominant regional source of Fe, Mn, and Co to the South Atlantic Gyre. Similarity between the TE and $^{228}$Ra distributions suggests that the fluxes scale proportionally[38]. Given a Fe/$^{228}$Ra ratio of 0.1 pmol atom$^{-1}$, the off-shelf dFe-Flux from the Congo River plume into the South Atlantic Ocean is $6.8 \pm 2.3 \times 10^8$ mol year$^{-1}$ (or $138 \pm 51$ mol m$^{-2}$ year$^{-1}$). On a global scale, this represents 0.7–2.3% of the global sedimentary Fe flux ($2.7$–$8.9 \times 10^{10}$ mol year$^{-1}$)[3,41], or ~$40 \pm 15\%$ (based on the uncertainties of our estimate) of total dFe atmospheric deposition into the entire South Atlantic Ocean[42]. The similarity between dMn and dCo fluxes in the Congo-shelf-zone and off-shelf transect (Table 1) is consistent with their conservative behavior along the Congo River plume, likely because of slow Mn and Co oxidation[43,44] and Mn photo reduction in surface waters[45], which keeps Mn in solution and facilitates its off-shelf transport. Our total off-shelf TE fluxes are similar to those reported elsewhere for other ocean margin regions[37,38], although in the current study this flux occurs over a much smaller area.

Atmospheric deposition is thought to be an important source of dFe to the surface ocean, and the West African margin receives dust fluxes which are amongst the highest in the world[46]. So how does atmospheric deposition compare to lateral dFe supply from the Congo across this region? Dissolved Al is a useful tracer of recent dust deposition to the surface ocean, and atmospheric deposition has been estimated from GA08 Al data[47] as 2.65 g m$^{-2}$ year$^{-1}$ for the offshore (3°S) section and 2.67 g m$^{-2}$ year$^{-1}$ for the coastal transect. The offshore values exclude stations within the coastal shelf zone where other dAl sources, certainly including a contribution from direct Congo River discharge, preclude the use of this tracer. Nevertheless, if all dAl within the study region were attributed to atmospheric deposition it would correspond to deposition of 26.2 g m$^{-2}$ year$^{-1}$[47]. The fractional composition and solubility of Fe in dust vary widely. Using a broad range of plausible dust Fe content (1.9–5.0%) and Fe solubility (0.14–21%)[48,49] suggests that Fe deposition in these regions is within the range of 13–4900 μmol m$^{-2}$ year$^{-1}$ (a minimum and maximum limit given the contribution of dAl from nonatmospheric sources in this zone), or $0.01$–$4.93 \times 10^7$ mol year$^{-1}$, 1–3 orders of magnitude lower than our estimated dFe fluxes from the Congo-shelf-zone to the same region (Table 1).

Whilst considerable in a global context, these estimates of atmospheric deposition therefore represent only a small fraction (<1%) of the TE fluxes into the Congo-shelf-zone. Combined with the strong correlation between TE concentrations, $^{228}$Ra concentrations and salinity, this strongly suggests that outflow from the Congo-shelf-zone dominates TE supply across this region, with atmospheric deposition only a minor contributing factor.

As the Congo is the dominant river in the region (approximately four times larger than other more northerly rivers combined[31]), we consider that the contribution of other rivers to the low salinity plume observed in the study region is minimal. The enhanced dFe-flux from the Congo, relative to other rivers (see Fig. 3), into the South Atlantic may partially result from stabilization by organic ligands[50]. However, elevated dissolved organic carbon (DOC) concentrations are common in many major river systems[51,52], and dFe removal in the Congo estuary has been explicitly demonstrated[2]. Therefore high DOC alone cannot explain the unique dFe distribution observed along the Congo plume. Similarly, near-conservative behavior of Fe has been found in some estuaries with rapid flushing[53]. Indeed, rapid lateral advection appears to enhance TE transport from the Congo compared with other rivers (Fig. 3). A similar feature has been reported in the Arctic Ocean, where rapid transport of river-derived organic carbon and TEs through the Transpolar Drift leads to nanomolar TE concentrations in the central Arctic Ocean[54].

The elevated dFe export observed here appears to impact phytoplankton in the South Atlantic Gyre. Primary production across extensive regions of the SE Atlantic is proximally limited, or co-limited by availability of the micronutrients Fe and Co due to limited atmospheric supply beyond the equatorial dust belt[10]. However, primary productivity within the offshore region in the current study was instead found to be limited by nitrogen availability[10], likely due to the large TE input from the Congo plume. Changes to the spatial orientation of the plume due to shifts in wind patterns or changing freshwater discharge may therefore directly affect TE supply and thus offshore primary production within the South Atlantic Gyre. Wind speeds are likely to decrease in the Congo region over the coming century[55] and a future reduction in Atlantic thermohaline circulation is projected to further alter prevailing wind patterns in the intertropical convergence zone[56,57]. On a decadal timescale, total annual rainfall across the Congo River Basin is not expected to change significantly, but an amplification of the seasonal variability of Congo River runoff is predicted[58]. There is therefore clear potential for consequences of climate change in the Congo region to affect nutrient availability and marine primary production in the SE Atlantic Ocean.

## Methods

**Sample collection and analysis.** Surface seawater samples (3 m depth) for Ra isotopes and dFe analyses were collected onboard R/V Meteor during the GEO-TRACES GA08 cruise in the Southern Atlantic between November 22 and December 27, 2015 (Fig. 1).

*Radium isotopes.* Surface samples (3 m depth) were collected by pumping ca. 250 L of seawater into a barrel. Seawater was then filtered through MnO$_2$-impregnated acrylic fiber (Mn-fibers) at a flow rate <1 L min$^{-1}$ to quantitatively extract Ra isotopes. After collection, the Mn-fibers were rinsed and air dried. Concentrations of $^{224}$Ra were determined using four Ra delayed coincidence counters (RaDeCC)[59]. The fibers were counted onboard and aged for 6 weeks, in order to allow excess $^{224}$Ra to completely decay. They were then recounted to determine $^{228}$Th concentrations and thus correct the total $^{224}$Ra for the supported activity. RaDeCC counters were calibrated with International Atomic Energy Agency (IAEA) reference solutions[60].

After measurement of $^{224}$Ra, fibers were ashed and subsequently leached in order to determine the activity of long-lived Ra ($^{228}$Ra and $^{226}$Ra) isotopes using a

high-purity, well-type germanium (HPGe) gamma spectrometer. As the remaining amount of ash obtained was too large to fit inside the well of the HPGe detector (Canberra Eurisys GMBH, EGPC 150), the ashes were subsequently leached followed by coprecipitation with $BaSO_4$. Ashing the fibers before leaching produced a more homogeneous material that was easier to handle as has been demonstrated occasionally elsewhere[61]. The fibers were ashed at 600 °C for 20 h, then leached in 6 M HCl followed by coprecipitation with $BaSO_4$. The precipitate was then sealed in 1 mL vials and analyzed after at least 3 weeks to allow $^{222}Rn$ to reach equilibrium with its parent $^{226}Ra$. Radium-226 concentrations were determined using the $^{214}Pb$ peak (352 keV) and the $^{214}Bi$ peak (609 keV), and $^{228}Ra$ concentrations were determined using the $^{228}Ac$ peaks (338 and 911 keV). Sample counting efficiencies were determined by spiking Mn-fibers with known amounts of $^{228}Ra$ and $^{226}Ra$, and processing similar to samples. Sample activities were corrected for detector background counts and fiber blank activities. The radium calibration solution was provided by the IAEA, and had a reported activity accuracy of 6% for $^{226}Ra$ and 5% for $^{228}Ra$. Measured precisions for $^{228}Ra$ and $^{226}Ra$ were ~5% (1–σ). These levels of accuracy and precision led to an uncertainty on the sample concentrations of <10%.

*Trace elements.* Surface trace element sampling was conducted using a tow fish deployed alongside the ship at about 3–4 m depth. Seawater was collected in a shipboard clean laboratory container and stored in acid-cleaned low density polyethylene 125 mL bottles. Samples for dTEs were collected using a cartridge filter (0.8/0.2 μm, Acropak 500 – PALL). All seawater samples were acidified onboard with ultra clean HCl (UpA grade, Romil) to pH 1.9. Freshwater Congo River samples were collected at three timepoints (April, July, and October 2017), retained for analysis of dFe after syringe filtration (0.20 μm, Millipore), and acidified as per seawater. Samples with concentrations below 20 nM dFe or dMn were measured following ref. [62]. Samples with higher concentrations were analyzed by inductively coupled plasma-mass spectrometry after dilution with ultra-pure 1 M $HNO_3$ (Romil SpA grade, sub-boiled), and calibration by standard addition. The accuracy and precision of measurements were evaluated by analysis of SAFe S, SAFe D2, and CASS6 reference seawater (Supplementary Table 2).

*Radium-228 inventory and trace element flux estimates.* The $^{228}Ra$ Congo-shelf-endmember was determined by the average $^{228}Ra$ concentrations (14.5 ± 3.5 dpm 100 L$^{-1}$; Table 1) of samples with salinity less than 32 psu, which best reflect river influenced waters in the Congo-shelf-zone. This excess $^{228}Ra$ was then corrected for the average offshore $^{228}Ra$ concentration (1.8 ± 0.5 dpm 100 L$^{-1}$)[63], in order to account for mixing with offshore waters containing a background concentration of $^{228}Ra$. The $^{228}Ra$ Congo-shelf-endmember is therefore 12.7 ± 3.6 dpm 100 L$^{-1}$. We estimated the $^{228}Ra$ inventory ($I_{228}$) in the surface water of our Congo-shelf-endmember as 2.8 ± 0.8 × 10$^9$ atoms m$^{-2}$ by considering a plume thickness of 5 m in the Congo-shelf-zone (Supplementary Fig. 4) and assuming a uniform $^{228}Ra$ distribution within this plume thickness. Radium-228 flux from the Congo-shelf-zone was determined using the $^{228}Ra$ inventory of our Congo-shelf-endmember and the residence time of water in the Congo-shelf-zone (Flux $^{228}Ra$ = inventory/residence time). A residence time of 3 days[13] is consistent with the Congo discharge volume required to produce the observed salinity within the plume region[30]. As our Congo-shelf-endmember is ~100 km away from the river mouth, the area of our sampling region was considered as a square of 100 × 100 km (whenever referring to the Congo-shelf-zone). Taking the $^{228}Ra$ at the lowest measured salinities ($S < 32$) as representative of the entire Congo River mouth zone likely produces a lower estimate of the true inventory because concentrations are likely higher at mid-salinities nearer to the river mouth and shoreline (Supplementary Fig. 3). Note that the regional satellite-derived salinity shown in Supplementary Fig. 5 corresponds only qualitatively with measured salinity for the Ra samples in the Congo-shelf-zone (~30 psu). Radium-228 flux from the Congo River was estimated by extrapolating the regression line to zero salinity (Fig. 2b) and multiplying the intercept by river discharge. These two approaches were used to check if the Congo River flux and the $^{228}Ra$ flux estimated for the Congo-shelf-zone were comparable, and thus to investigate to what extent the River flux on its own could explain $^{228}Ra$ concentrations on the shelf.

Because dFe, dMn, and dCo in our study appear to have a source similar to Ra, the $^{228}Ra$ flux was used to determine the fluxes of these TEs in the Congo-shelf-zone, by multiplying the $^{228}Ra$ flux in this region by the average ratio of dTE concentrations (Table 1) observed in samples with the lowest measured salinity (i.e., <29 psu; TE Congo-shelf-endmember) and the $^{228}Ra$ concentration of the Congo-shelf-endmember (dTE/$^{228}Ra$). Samples with salinity < 29 represent intermediate TE endmembers that reflect mixing between the river and seawater ($S > 33$ psu). Ra and TE sampling locations do not coincide exactly, as unlike Ra samples, TE samples were collected as small volumes (125 mL) from a tow fish while the research vessel was underway.

The residence time of 7 ± 2 days at the start of the off-shelf transect (3°S; between stations 1218 and 1229, at the shelf break) was determined using the $^{224}Ra/^{228}Ra$ ratios as per ref. [64]. We used the $^{224}Ra/^{228}Ra$ ratio at station 1218 as our initial ratio. Radium-224 was detectable over the next 40 km at the next two stations along this transect (station 1229). Once Ra isotopes are released into the water column, their activities decrease with increasing distance from the source as a result of dilution and radioactive decay. Both isotopes are affected by dilution, but

radioactive decay is negligible for $^{228}Ra$ (half-life = 5.8 years) over short distances, so changes in the $^{224}Ra/^{228}Ra$ ratio reflect the time elapsed since the water was isolated from the source. Within the Congo plume, strong density stratification isolates the freshwater plume from bottom waters, so surface waters are unlikely to be affected by additional Ra input. Therefore, the residence time ($T$) can be derived as follows:

$$T = \ln \left[ \frac{\left( ^{224}Ra/^{228}Ra \right)_i}{\left( ^{224}Ra/^{228}Ra \right)_o} \right] \times \frac{1}{\lambda_{224}}, \quad (1)$$

where $(^{224}Ra/^{228}Ra)_i$ is the initial ratio at station 1218, $(^{224}Ra/^{228}Ra)_o$ is the ratio observed away from the source (offshore) at station 1229, and $\lambda_{224}$ is the decay constant of $^{224}Ra$.

Previous studies have combined the $^{228}Ra$ flux with water column dTE to $^{228}Ra$ ratios (TE/$^{228}Ra$) in order to quantify shelf-ocean input rates[37,38]. We propose the use of this approach to estimate the fluxes of TEs (dFe, dMn, and dCo) from the Congo River plume to the Atlantic Ocean following ref. [38].

$$Flux\ TE = Flux\ Ra \times \left[ \frac{dTE_{shelf} - dTE_{offshelf}}{Ra_{shelf} - Ra_{offshelf}} \right], \quad (2)$$

where $dTE_{shelf}$ and $^{228}Ra_{shelf}$ are the average concentrations of the dTE and $^{228}Ra$ in surface waters over the shelf (between station 1218 and station 1229), respectively; and $dTE_{off-shelf}$ and $^{228}Ra_{off-shelf}$ are the dTE concentration and $^{228}Ra$ concentrations in surface waters of the open ocean station (station 1234). Note that a diffusion-dominated system between the stations 1218 and 1234 was observed as indicated by a linear gradient in both dFe and $^{228}Ra$ distributions. The dTE/Ra ratios are presented in Table 1.

**Reporting summary.** Further information on research design is available in the Nature Research Reporting Summary linked to this article.

## Data availability
All data are available in the main text or the Supplementary Information. Source data for dTEs and Ra isotopes are provided as a Source Data File.

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

## Acknowledgements

We thank the captain and crew of the RV Meteor M121 cruise/GEOTRACES
GA08 section and the chief scientist M. Frank for cruise support. We thank S. Koesling,
P. Lodeiro, and C. Schlosser for their assistance in sample collection on the GA08
expedition and T. Browning for the assistance with satellite data. We thank Mr Bomba
Sangolay from the National Institute of Fisheries Research (Instituto Nacional de
Investigação Pesqueira, Luanda, Angola) for collection of samples in the Congo River.
The PhD Fellowships to L. H. Vieira and S. Krisch were funded by the Conselho
Nacional de Desenvolvimento Científico e Tecnológico, Brazil (CNPq - grant number
239548/2013-2) and the Deutsche Forschungsgemeinschaft (AC 217/1-1), respectively.
The cruise was funded by the Deutsche Forschungsgemeinschaft.

## Author contributions

E.P.A., J.S., and L.H.V. designed the GA08 study. L.H.V. carried out the sampling, analyzed
the radium samples, and wrote the first draft of the manuscript. A.J.B., M.J.H. and L.H.V.
worked on subsequent drafts. S.K. and M.J.H. analyzed GA08 trace element samples. A.J.B.
and M.J.H. contributed with interpretation of results. V.L. helped with Ra analysis by
gamma spectrometry. All authors contributed to final version of the manuscript.

## Competing interests

The authors declare no competing interests.

## Additional information

**Peer review information** *Nature Communications* thanks Jordon Beckler and the other,
anonymous, reviewer(s) for their contribution to the peer review of this work. Peer
reviewer reports are available.

**Publisher's note** Springer Nature remains neutral with regard to jurisdictional claims in
published maps and institutional affiliations.

