## [Peer Review File · Nature Communications]

Reviewers' comments:

Reviewer #1 (Remarks to the Author):

NCOMMS-19-18100 review

This paper describes a set of measurements of salinity, dissolved Fe and Ra isotopes to quantify the input of dFe from the River Congo to the SE Atlantic Ocean. The data seem to be quite accurate, as demonstrated by their analysis of the GEOTRACES consensus reference seawater samples (and other CRMs). The text is well written and their logic towards calculating the dFe flux is easy to follow. However, there is nothing in the paper about seasonality (in all aspects: river flow rate, river concentrations, coastal mixing, offshore transport, etc.). These factors all affect the fluxes that are calculated, and they are all based on a single set of samples collected over a short period of time. Including additional variance due to seasonality would enlarge the relative standard deviations of the fluxes. This should be addressed.

I have only a few specific comments, keyed to line number:

127-128: This last sentence is circular logic. The previous sentence assumes conservative mixing to yield an effective riverine end-member for Ra-228, so it could simply be stated that this assumes no additional inputs of Ra-228.

130-138: This text is unnecessary. It is sufficient to say what has been reported for other rivers (around 20 dpm/100L), and that the apparent riverine end-member for Ra-228 (85 dpm/100L on line 124, but 79 dpm/100L on line 140!!) is much higher than that. Then list the possible reasons why, starting on line 138. Can you decide which reason is most likely? Why do you state two different end-member concentrations? There is no value in calculating atom fluxes that would be expected if the river were closer to 20 dpm/100L.

143-147: Remind the readers that these TE fluxes are based on the unusually high Ra-228 flux (because of the unusually high riverine end-member concentration) and include some text about how seasonal variations would affect these fluxes.

164: Explain how aerosol Fe solubility was measured or estimated. Include an error analysis on the statement that the dFe flux is 40% of the atmospheric flux. I get 40+/-20%. If you add the Congo river dFe flux to the atmospheric flux, then the Congo flux is 29+/-13% of the total dFe flux to the entire South Atlantic. That is still significant, but the error propagation should be included.

217-218: This sentence is flawed. Do you mean the accuracy was within 6% and 5% of the certified values?

226: Please state the concentration of added HCl. If you added 2 mL of 12M HCl per liter, then you added +0.024M HCl (pH 1.9).

Reviewer #2 (Remarks to the Author):

The manuscript entitled "Unprecedented Fe delivery by the River Congo to the South Atlantic Gyre" by Vieira et al. synthesizes important results regarding the role of the Congo River in delivering iron, radium, and other trace elements to the Atlantic Ocean. The authors demonstrate that river discharge, river-dominated coastal sediments, or submarine groundwater discharge in a river-dominated area combine to ensure that globally, the Congo is the most significant riverine source of iron to the oceans. The topic of the paper is of high interest to the majority of chemical oceanographers engaged in modern research, as the debate surrounding the sources of iron to the ocean is one of the most currently contested. The authors employ state of the art techniques and the work is compliant with GEOTRACES – going a long way in ensuring that analyses and quality control was likely meticulous and sufficient. I do have some questions regarding the data interpretation in several areas of the manuscript, but am excited with the larger implications of the paper, which demonstrate the river is important with respect to synoptic scale ocean processes (alleviation of Fe-limitation in the Western Atlantic), but also mechanistically speaking (that an organic/Fe rich river system may provide a disproportionately large contribution of Fe to the oceans). As far as relative importance in the field, rivers have been discounted relative to

atmospheric dust, and more recently sediments and hydrothermal vents. Much of this stems from early assumptions that iron mostly flocculates and is therefore not worth considering on global scales in models, as well as the fact that the "world average river" from which estimates are primarily derived do not consider smaller rivers that may drain more organic/Fe-rich coastal plain sediments. For this reason, the number of studies focusing on river sources of Fe have been relatively limited in number in the last couple of decades (in my opinion). It is nice to see a GEOTRACES data set that focuses specifically on riverine discharge, which to my knowledge have also been limited beyond at least the Amazon (nothing yet for the Mississippi River for example). Sure enough, the results illustrate the river is important – with observed concentrations far exceeding expectations based on river flow alone – with the overall conclusions that rivers cannot be ignored in global ocean iron budgets.

I provide comments below, with moderate (albeit thought-provoking comments) comments/concerns/areas for improvement addressed by line number. I recommend publishing after consideration of the these points.

1. The title, "Unprecedented Fe delivery by the River Congo to the South Atlantic Gyre" is a bit misleading in that while the river is ultimately responsible for the Fe delivery to the gyre, it is likely first delivered to the sediments first, so there is this physical disconnect. I don't have the perfect title either, but something that conveys the idea: "...by the River Congo-dominated ocean margin to the..."

Overall, I think the authors should approach this as a point of pride - they are demonstrating that river plumes can influence larger scale oceanographic processes if not directly, then through a continental margin "intermediate", that receives TE and is then culprit in supporting fluxes to the water column.

2. Line 17: I would change to "The Congo dFe outflow therefore contributes significantly to relieving Fe-limitation...", because there is no data in the manuscript that demonstrates the river is the sole or primary source of Fe to the South Atlantic Gyre, as it currently (could) read.

3. Line 29: Is this 3% figure, for the riverine delivery of new ocean Fe, obtained before or after accounting for flocculation in estuaries? Ambiguous language in the literature on this topic have led to a lot of inconsistencies in the community, so I recommend clarifying for the reader.

4. Line 33: "Unique" is an adjective describing the equatorial plume itself, but the conditions in the first half of the same sentence refer to conditions about the river and the shelf. How is the plume itself unique? I think this is too vague.

5. Lines 40-42: Please add Radium references here. My own question is whether or not we would expect to see additional Radium released from continental margin sediments hosting mineral dissolution processes, e.g. microbial iron reduction? Later on in the paper (line 140) you seem to suggest this is possible. So, I am concluding yes, but would appreciate some more details as it is critical to interpreting the work.

6. Line 45: Please clarify this figure and mark very clearly, on the diagram itself perhaps, the 3 groups of samples that are discussed: the "Congo Shelf Zone", the "Coastal Transect", and the "Offshore Plume". While I think all the info can be gathered from the text as it stands, it did take me a couple of reads to really nail these down.

7. Line 48: Add "offshore" before "3 degrees S", and perhaps one line that says this was the seasonal high-flow period.

8. Line 67-78: I would move a version of the sentence "low removal primarily reflects..." to the beginning of this paragraph (e.g. before "Extrapolating" in line 57), because otherwise it appears as if you are trying to state as the primary point of this section that only 50% of the iron is removed. It sets the reader up for confusion.

9. Line 59: Saying that you are extrapolating to a "zero salinity endmember" is also confusing, because this would generate a that technically occurs prior to any estuarine flocculation processes (that happen at salinities > 0). Instead, extrapolation of the data points in Figure 2A actually

reveals the estuarine endmember that should have already undergone mixing and hosted flocculation processes. I would support language, "extrapolating to an estuarine endmember"

10. Line 74: Are these ratios specific to the Congo? Please state if so. Otherwise, I don't think this is really a valid exercise given how variable river concentrations can be.

11. Line 75: Add "relative to seawater background" before "in the Congo"?

12. Line 78 and others: I went to the cited reference because I was questioning the logic here, and I don't think the cited paper supports the argument in this manuscript: "The Congo River plume may be classified as surface-advected (Yankovsky & Chapman, 1997); buoyant river inflow remains on top of shelf water forming a thin layer that, decoupled from bottom stress, spreads many hundreds of kilometres offshore." Now, just because turbulent benthic interactions are minimal offshelf, this doesn't mean upwelling cannot be significant off-shelf. In fact, the cited paper proceeds to state in the next paragraph: "An estuarine type circulation has been identified here, where surface outflow is compensated at depth by a residual up-canyon transport of saline water (Denamiel, Budgell, & Toumi, 2013; Eisma & Van Bennekom, 1978)." I would remove or rephrase line 78 – I don't think it supports the argument.

Here is where I hope to ignite some thought: the Congo River is the only large river in the world still connected to its deep sea fan via a turbidite canyon, and because of these turbidity flows, the Congolobe expedition observed incredibly high rates of accumulation of organic/iron minerals (>2 cm/year) and high rates of iron solubilization from shelf-break down to deep-sea fan sediments (consistent with those in Fe-rich estuaries) up to 800 km offshore (Beckler et al. 2016; Taillefert et al. 2017; Rabouille et al. 2017 and other references therein). Under conditions of intense iron-diagenesis in sediments, a corresponding flux of Fe and TE typically also occurs, so I suspect that these processes, i.e. upwelled slope/deep-sea waters previously in contact with these sediments, may also be contributing to plume enrichments (in addition to shelf sediments as stated in line 140). The actual offshore transect in this manuscript processed to the North of the Congo River canyon. I wonder if during other times of year when the plume may be situated more southward, if we may expect even a higher contribution of sediment-derived Fe within the plume, and overall contributing to the gyre?

13. Some comments/alternative explanations relating to differences in Fe content, but similar salinities, for the CSZ and off-shelf sample sets (Figure 2):

a. Could the northern freshwater plumes be from a different river source and thus contain a different amount of Fe altogether?

b. Are the northern and southern sample plume water masses both from the Congo but from different time periods, as weekly variations in flow may result in various concentrations of iron in the river water?

14. Line 99: I would remove this comparison, because the utility of the "zero salinity" end-member is virtually meaningless now that you have established that the dominant factor is indeed shelf input. This really had me confused.

15. Line 140: I suspect based on other thoughts throughout the paper that you believe here that explanation (ii) is the most logical. I do feel the atypical sedimentary environment of the Congo margin can explain this phenomenon.

-Jordon Beckler

Reviewer #3 (Remarks to the Author):

Review "Unprecedented Fe delivery by the River Congo to the South Atlantic Gyre" by L.H. Vieira

This study estimates fluxes of dissolved trace elements (Fe, Co, and Mn) based on ²²⁸Ra data from the Congo shelf. The results of this study will be useful for the scientific community; however, I think that the paper needs to be significantly improved before considering it for publication. I am also wondering if Nature Communication Journal is the right place for this study, because the method applied is not new, the authors presented a flux of trace elements from the

Congo River that is new information but not original, and the short length of this Journal seemed to have prevented the authors to completely discuss the results. I have three main points that I think the authors should address:

1) I noted some odd points in the method that I describe in details below. One point that needs to be addressed on the method is that the authors did not correct the ^{228}Ra average activity in the Congo-shelf-zone from offshore ^{228}Ra activities (if they did, it is not clearly written). This would affect the ^{228}Ra flux, which is then used for the TE fluxes. Therefore, the fluxes of the shelf would likely need to be re-calculated.

2) More comparisons are needed regarding the fluxes of Ra^{228} , Fe, Co, and Mn. An estimation of dust deposition should be included, especially after the recent study published by Mendez Barraqueta et al. 2019 based on data from the same cruise showing significant atmospheric deposition. The high dFe concentration on the Congo shelf might well be the results of the combination of high dust deposition, shelf sediment inputs, and large Congo River, rather than attributing the Fe in seawater solely to the Congo River inputs. Comparison of these fluxes with other study areas should also be included to answer the question: Is the Congo shelf a unique supplier of TE to the ocean?

3) In my opinion, the title, abstract, and intro needs to be re-written a little. I personally don't like the catchy Nature forced title "unprecedented Fe delivery". An alternative title could be in the lines of "the Congo shelf represents a unique (or major) supplier of TE to the South Atl..." I am confused by the title/Abstract/intro that focus on the Congo River. The TE fluxes are estimated from Ra^{228} , which is a sediment tracer (except if the Congo river has exceptionally high ^{228}Ra activity, which hasn't been demonstrated in this study). The fluxes are estimated using Ra for the "Congo-shelf-zone" and on an "off-shelf transect", which include inputs from both the river and from sediment. The salinity range is pretty high, mainly above 32, which suggest less than 20% of freshwater contribution in the study zone. The author estimates the Congo river flux (discharge times concentration in the river), which is not necessarily comparable to the Ra flux because the latter is estimated for a specific area and not all the river freshwater discharge actually makes it into this restricted zone. I think there are a lot of discussion that has been cut probably because of the very short format of the journal, but I feel that it would improve the paper to provide more details regarding what these fluxes actually represent.

Methods:

- The second count for Ra^{224} was performed 6 weeks later whereas the second count for Th^{228} is usually done 2 weeks after collection. Was it really done after 6 weeks? Then some Bateman correction should probably be applied.

- After measurements of short-lived on the RaDeCc , the authors said that the fibers were ashed and then leached. Did the author leached the ashes? I am aware of two methods: ash (Charette et al. 2001) or leach/co-precipitate (Moore method) the fibers but I have never seen both processes applied one after the one. Is it a typo by the authors or was it the real procedure?

- The Congo-shelf-zone flux of ^{228}Ra was estimated as inventory/residence time. The authors used the excess ^{228}Ra in the flux calculation. The excess Ra^{228} should be the average ^{228}Ra concentration in the study area corrected from the offshore ^{228}Ra concentration. This is to account for mixing with offshore waters that have a background ^{228}Ra . If the authors had used ^{224}Ra , there would be no need to consider the excess ^{224}Ra because offshore waters have negligible amount of ^{224}Ra . The authors should use an excess ^{228}Ra of $14.5 - 4 = 10.5$ based on figure 2, the ^{228}Ra is about 4 dpm/100L outside of the Congo shelf zone. Or 8.6 dpm/100L considering the average offshore activities from the offshelf transect on table 1, it depends on what this offshore average value is. This will change the fluxes of TE... Cf Moore, W. S.:

Inappropriate attempts to use distributions of ^{228}Ra and ^{226}Ra in coastal waters to model mixing and advection rates, *Continental Shelf Research*, 105, 95–100, doi:10.1016/j.csr.2015.05.014, 2015.

Other comments:

- What is the input of atmospheric deposition of Fe to the study area? What is the contribution of dust the high Fe concentration in the Congo River plume? Dust is a significant source of Fe to the

ocean (Jickells et al., 2005), therefore, the contribution of dust-derived Fe in the surface concentration cannot be ignored. Especially considering a recent publication from Menzel Barraqueta et al. (2019) showing that the study area receives one of the highest atmospheric deposition rate to the Atlantic Ocean calculated from Al data from the same cruise GA08 (~8-10 g/m²/y). A rough estimate of the flux of Fe from dust can be calculated using the average atm. deposition, the concentration of Fe if measured on aerosol during the cruise or the composition of Fe in crust, and the solubility factor. Similar calculations can be done for Mn, and Co shown in the suppl. info. A more precise location can also be done since the deposition rate is estimated for the same stations. The author could thus perform a mass balance and potentially correct the Fe concentration that is due to dust.

- The comparison of the Fe:Mn ratio in the river to the Fe:Mn ratio in the shelf-zone sounds a bit off to me, but I might be wrong. The lower Fe:Mn ratio in the shelf could simply reflect a different behavior of Fe and Mn. Fe is removed faster than Mn, thus the faster removal of Fe would result in a lower Fe:Mn ratio in the shelf compare to the Fe:Mn ratio in river without involving additional source as suggested by the authors.

- Can the authors discuss the three hypothesis that could explain the high Ra flux from the Congo-shelf-zone. 1) exceptionally high Ra in the Congo, 2) high Ra diffusion from sediment, 3) SGD. Which one is the most likely? Why Ra would be much higher in the Congo River compare to other riverine system.

- How does the Ra and TE fluxes from the Congo shelf zone compared to fluxes estimated in other regions? This could be compared to the TE fluxes to the north Atlantic Ocean from Charette, M. A., Lam, P. J., Lohan, M. C., Kwon, E. Y., Hatje, V., Jeandel, C., Shiller, A. M., Cutter, G. A., Thomas, A., Boyd, P. W., Homoky, W. B., Milne, A., Thomas, H., Andersson, P. S., Porcelli, D., Tanaka, T., Geibert, W., Dehairs, F. and Garcia-Orellana, J.: Coastal ocean and shelf-sea biogeochemical cycling of trace elements and isotopes: lessons learned from GEOTRACES, *Phil. Trans. R. Soc. A*, 374(2081), 20160076, doi:10.1098/rsta.2016.0076, 2016.

- Another approach to estimate the Congo river input could be used. The authors used the TE concentrations measured in the river and the river discharge. An estimation of the amount of freshwater actually entering the Congo-shelf zone can be done using the average salinity and applying conservative mixing of salinity to calculate the volume of freshwater instead of using the river discharge. Doing so, the source of TE from the freshwater Congo river will be directly comparable to the sediment derived estimate of Ra (since the mol/y unit implies a specific surface considered).

- The authors ruled out the benthic supplies of Ra and TE because of the shallow Congo river plume. But we don't really know what is the water column depth along the transect. Since the plume is moving along shore, it could be hugging the coast in shallow water. Thus, it would be useful to know the water column depth compare to the 15 m of river plume. Similarly, the odv plot figure 4 in suppl. info has no bathymetry, how steep is the shelf? Maybe adding the bathymetry to the Figure 1 would already give some information.

- Can the author discuss the difference of Fe flux between the Congo shelf zone and the off shelf transect? Does this provide information on the scavenging rate or biological uptake of Fe in between the two zones?

- Can the author discuss the exceptionally high dFe concentration 600 km from the Congo river mouth? Is the Fe 600 km away from the mouth river from the river or does it has another origin? That is just a side comment that I am curious about.

Comment on figures:

- Figure 1: I would add a large scale map of Africa for example in a subplot to better situate the study area, to be more visual. Add the units for latitude and longitude. Could the authors show the salinity underway data to better visualize the actual plume during the survey? Should be better than satellite image, especially because the authors acknowledge a difference in salinity between satellite and in situ data. Add the bathymetry to see the width and steepness of the shelf.
- In general, I think that surface plots would be useful and could be added as subplots to the figure 2 for example since the cruise track is composed by an along shore transect and an offshore transect.
- Figure 2: switch panels c and d to show Fe in a) and c), and Ra in b) and d)
- Figure 3: the Congo data do not stick out, I would exchange the symbol between Congo and Mississippi to highlight the data from this study in red. Based on this graph, the ^{228}Ra are consistent with other estuaries. Could this help in the discussion of the 3 hypotheses about the high flux of ^{228}Ra I mentioned earlier?

We would like to thank the reviewers for their constructive comments and suggestions. We followed the majority of the reviewers' suggestions as described below. Our responses to the reviewers' comments are written in italic, and transcriptions from the revised manuscript are in bold with initial lines in brackets.

Reviewers' comments:

Reviewer #1

NCOMMS-19-18100 review

This paper describes a set of measurements of salinity, dissolved Fe and Ra isotopes to quantify the input of dFe from the River Congo to the SE Atlantic Ocean. The data seem to be quite accurate, as demonstrated by their analysis of the GEOTRACES consensus reference seawater samples (and other CRMs). The text is well written and their logic towards calculating the dFe flux is easy to follow. However, there is nothing in the paper about seasonality (in all aspects: river flow rate, river concentrations, coastal mixing, offshore transport, etc.). These factors all affect the fluxes that are calculated, and they are all based on a single set of samples collected over a short period of time. Including additional variance due to seasonality would enlarge the relative standard deviations of the fluxes. This should be addressed.

Thank you for the suggestion. A discussion about seasonal changes has been added to the revised manuscript as follows:

[Line 143] ***“The large ²²⁸Ra and TE fluxes determined here are derived from their large inventories in the Congo-shelf-zone, and do not take into account seasonal variations in their supply to surface waters. Sample collection occurred during the high discharge season of the Congo, but seasonal variations of the river flow (mean annual range between 35 and 60 x 10³ m³ s⁻¹ ³⁹ with a ca. twofold seasonal variation in dFe concentrations (Supplementary Table 1)) can strongly affect the Congo River plume dispersion⁴⁰, and potentially the delivery of river-derived materials to the SE Atlantic Ocean. The seasonal variation in benthic supply to***

overlying plume waters of ^{228}Ra and TEs on the Congo shelf is unconstrained.” References in the revised manuscript.

I have only a few specific comments, keyed to line number:

127-128: This last sentence is circular logic. The previous sentence assumes conservative mixing to yield an effective riverine end-member for Ra-228, so it could simply be stated that this assumes no additional inputs of Ra-228.

Confusing wording has been revised. See the changes in the revised manuscript:

[Line 115] ***“Conservative mixing between the Congo-shelf-endmember and offshore waters (Fig. 2b) indicates a riverine ^{228}Ra effective-zero-salinity-endmember concentration of 85 ± 4 dpm 100 L^{-1} , including both dissolved and desorbed Ra. Together with the river discharge ($1.3 \times 10^{12}\text{ m}^3\text{ yr}^{-1}$)³¹, this suggests a fluvial ^{228}Ra flux of $4.8 \pm 0.4 \times 10^{21}$ atoms yr^{-1} , which is similar to the ^{228}Ra flux estimated for the Congo-shelf-zone ($3.4 \pm 0.9 \times 10^{21}$ atoms yr^{-1}). If the assumption of conservative mixing behavior is correct, the effective-zero-salinity-endmember would be similar to actual river Ra concentrations, which are not known.”***

130-138: This text is unnecessary. It is sufficient to say what has been reported for other rivers (around 20 dpm/100L), and that the apparent riverine end-member for Ra-228 (85 dpm/100L on line 124, but 79 dpm/100L on line 140!!) is much higher than that. Then list the possible reasons why, starting on line 138. Can you decide which reason is most likely? Why do you state two different end-member concentrations? There is no value in calculating atom fluxes that would be expected if the river were closer to 20 dpm/100L.

The activity of 85 dpm/100 L takes into account the desorbed and dissolved Ra fractions, while 79 dpm/100 L corresponds to only the dissolved Ra, estimated by the difference between the total activity (85 dpm/100 L) and the desorbed Ra activity (6 dpm/100 L). By doing this, we can compare the potential dissolved Ra activity in the Congo River with other systems (~20 dpm/100 L).

The most likely explanation cannot be number (i), because rivers do not show large variation in Ra activity, and the Ra concentration in the Congo would have to be higher by factor of 4 to

support the observed flux. Numbers (ii) and (iii), or a combination of both, may be more likely.²²⁸ *Ra* diffusion from shelf sediments has a large variability (e.g. Vieira et al., 2019 (doi.org/10.1016/j.marchem.2018.11.001)) and the existence of another source of *Ra* such as submarine groundwater discharge in a large river system such as the Congo is rather possible (e.g., Moore, 1997, doi.org/10.1016/S0012-821X(97)00083-6). We do not add more information about this issue as that would only be speculation. These two mechanisms are challenging to distinguish based on what data we have from an understudied region. See the text in the revised manuscript:

[Line 132] ***This suggests that either (i) the dissolved ²²⁸Ra concentration in the Congo River is exceptionally high compared to other large rivers (~79 dpm 100 L⁻¹, vs <20 dpm 100 L⁻¹ elsewhere); (ii) ²²⁸Ra diffusion from shelf sediments in this region is anomalously high compared to other regions globally; or (iii) there is another source of *Ra* such as submarine groundwater discharge^{34, 35}. Based on observations elsewhere of large variability of ²²⁸Ra diffusion from shelf sediments³⁶ and SGD input¹⁴, (ii) and (iii), or a combination of both, are most likely.***

In addition, we must do this calculation to show that the Congo fluxes we estimate are unusual. The flux calculation using the global “typical” river value allows us to show that if the Congo River were like other major river systems, then the benthic fluxes would be unusually large.

143-147: Remind the readers that these TE fluxes are based on the unusually high *Ra*-228 flux (because of the unusually high riverine end-member concentration) and include some text about how seasonal variations would affect these fluxes.

*High *Ra* fluxes were due to the large inventory, as well as high *Fe* fluxes are due to high concentrations. A discussion about seasonal variability has been added, see comment above.*

164: Explain how aerosol *Fe* solubility was measured or estimated. Include an error analysis on the statement that the d*Fe* flux is 40% of the atmospheric flux. I get 40+/-20%. If you add the Congo river d*Fe* flux to the atmospheric flux, then the Congo flux is 29+/-13% of the total d*Fe* flux to the entire South Atlantic. That is still significant, but the error propagation should be included.

Atmospheric flux (14.3×10^8 mol/yr, with no stated uncertainties) was estimated based on data from Duce and Tindale, 1991 (doi.org/10.4319/lo.1991.36.8.1715). The uncertainty of 40% (25% to 55%) is added to the revised manuscript based on the uncertainties of our estimate. No comparison with “total” flux has been made because we do not have a complete budget. Nonetheless, we make a comparison with sedimentary flux reported elsewhere. See also our answer to reviewer 3 related to atmospheric deposition.

[Line 168] “*(...) approximately $40 \pm 15\%$ of the total dFe atmospheric deposition into the entire South Atlantic Ocean, based on the uncertainties of our estimate.*”

[Line 176] “*Atmospheric deposition is thought to be an important source of dFe to the surface ocean and the West African margin receives dust fluxes which are amongst the highest in the world. So how does atmospheric deposition compare to lateral dFe supply from the Congo across this region? Dissolved Al is a useful tracer of recent dust deposition to the surface ocean, and atmospheric deposition has been estimated from GA08 Al data⁴⁶ as $2.65 \text{ g m}^{-2} \text{ yr}^{-1}$ for the offshore (3° S) section and $2.67 \text{ g m}^{-2} \text{ yr}^{-1}$ for the coastal transect. The offshore values exclude stations within the coastal shelf zone where other dAl sources, certainly including a contribution from direct Congo discharge, preclude the use of this tracer. Nevertheless, if all dAl within the study region were attributed to atmospheric deposition it would correspond to deposition of $26.2 \text{ g m}^{-2} \text{ yr}^{-1}$ ⁴⁶. The fractional composition of dust and the elemental solubility within dust vary widely. Using a broad range of plausible dust Fe content (1.9-5.0%) and Fe solubility (0.14-21%)^{47, 48} suggests that Fe deposition in these regions is within the range of $13\text{-}4900 \mu\text{mol m}^{-2} \text{ yr}^{-1}$ (a minimum and maximum limit given the contribution of dAl from non-atmospheric sources in this zone), or $0.01\text{-}4.93 \times 10^7 \text{ mol yr}^{-1}$, 1-3 orders of magnitude lower than our estimated dFe fluxes from the Congo-shelf-zone to the same region (Table 1).*”

217-218: This sentence is flawed. Do you mean the accuracy was within 6% and 5% of the certified values?

We mean that the radium solution used to calibrate the detector was provided by the IAEA and had a reported activity accuracy of 6% for ^{226}Ra and 5% for ^{228}Ra . There is no SRM/CRM

available for ^{228}Ra to date. We changed “reference solution” to “calibration solution” in order to avoid confusion.

[Line 252] ***“The radium calibration solution was provided by the IAEA, and had a reported activity accuracy of 6% for ^{226}Ra and 5% for ^{228}Ra . Measured precisions for ^{228}Ra and ^{226}Ra were ~ 5% (1- σ). These levels of accuracy and precision led to an uncertainty on the sample concentrations of <10%.”***

226: Please state the concentration of added HCl. If you added 2 mL of 12M HCl per liter, then you added +0.024M HCl (pH 1.9).

The concentration of distilled HCl acid varies by batch, so we do not have the exact concentration, but the final pH of seawater samples was verified to be 1.9.

Reviewer #2

The manuscript entitled “Unprecedented Fe delivery by the River Congo to the South Atlantic Gyre” by Vieira et al. synthesizes important results regarding the role of the Congo River in delivering iron, radium, and other trace elements to the Atlantic Ocean. The authors demonstrate that river discharge, river-dominated coastal sediments, or submarine groundwater discharge in a river-dominated area combine to ensure that globally, the Congo is the most significant riverine source of iron to the oceans. The topic of the paper is of high interest to the majority of chemical oceanographers engaged in modern research, as the debate surrounding the sources of iron to the ocean is one of the most currently contested. The authors employ state of the art techniques and the work is compliant with GEOTRACES – going a long way in ensuring that analyses and quality control was likely meticulous and sufficient. I do have some questions regarding the data interpretation in several areas of the manuscript, but am excited with the larger implications of the paper, which demonstrate the river is important with respect to synoptic scale ocean processes (alleviation of Fe-limitation in the Western Atlantic), but also mechanistically speaking (that an organic/Fe rich river system may provide a disproportionately large contribution of Fe to the oceans). As far as relative importance in the field, rivers have been discounted relative to atmospheric dust, and more recently sediments and hydrothermal vents.

Much of this stems from early assumptions that iron mostly flocculates and is therefore not worth considering on global scales in models, as well as the fact that the “world average river” from which estimates are primarily derived do not consider smaller rivers that may drain more organic/Fe-rich coastal plain sediments. For this reason, the number of studies focusing on river sources of Fe have been relatively limited in number in the last couple of decades (in my opinion). It is nice to see a GEOTRACES data set that focuses specifically on riverine discharge, which to my knowledge have also been limited beyond at least the Amazon (nothing yet for the Mississippi River for example). Sure enough, the results illustrate the river is important – with observed concentrations far exceeding expectations based on river flow alone – with the overall conclusions that rivers cannot be ignored in global ocean iron budgets.

I provide comments below, with moderate (albeit thought-provoking comments) comments/concerns/areas for improvement addressed by line number. I recommend publishing after consideration of these points.

1. The title, “Unprecedented Fe delivery by the River Congo to the South Atlantic Gyre” is a bit misleading in that while the river is ultimately responsible for the Fe delivery to the gyre, it is likely first delivered to the sediments first, so there is this physical disconnect. I don’t have the perfect title either, but something that conveys the idea: “...by the River Congo-dominated ocean margin to the...”

Overall, I think the authors should approach this as a point of pride - they are demonstrating that river plumes can influence larger scale oceanographic processes if not directly, then through a continental margin "intermediate", that receives TE and is then culprit in supporting fluxes to the water column.

Thank you, we accepted the suggestion and the title has been changed. It now reads: “Unprecedented Fe delivery from the River Congo margin to the South Atlantic Gyre.”

2. Line 17: I would change to “The Congo dFe outflow therefore contributes significantly to relieving Fe-limitation...”, because there is no data in the manuscript that demonstrates the river is the sole or primary source of Fe to the South Atlantic Gyre, as it currently (could) read.

Changed as suggested:

[Line 16] ***“The River Congo therefore contributes significantly to relieving Fe-limitation of phytoplankton growth across much of the South Atlantic.”***

3. Line 29: Is this 3% figure, for the riverine delivery of new ocean Fe, obtained before or after accounting for flocculation in estuaries? Ambiguous language in the literature on this topic have led to a lot of inconsistencies in the community, so I recommend clarifying for the reader.

This 3% from Raiswell refers to after estuarine removal. Raiswell guesses something around 90% estuarine removal, although this actually varies widely from 60-99% (even within the Boyle 1977 reference usually cited as demonstrating 90% removal there is pronounced variation between rivers at different times) .

[Line 28] ***“Whilst riverine Fe concentrations are 3-5 orders of magnitude greater than those in seawater⁸, rivers provide only ~3% (after estuarine removal) of the new Fe delivered annually to the oceans⁹.”***

4. Line 33: “Unique” is an adjective describing the equatorial plume itself, but the conditions in the first half of the same sentence refer to conditions about the river and the shelf. How is the plume itself unique? I think this is too vague.

It is unique in the sense that it is the only major river system with a plume entering an into an eastern boundary upwelling region. The text has been changed for clarification in the revised manuscript:

[Line 33] ***“The Congo is the second largest river on Earth by discharge volume¹¹, and is the only major river to discharge into an eastern boundary ocean region with a narrow shelf¹²-unique characteristics for a near-equatorial river plume subject to low Coriolis forces¹¹.”***

5. Lines 40-42: Please add Radium references here. My own question is whether or not we would expect to see additional Radium released from continental margin sediments hosting mineral dissolution processes, e.g. microbial iron reduction? Later on in the paper (line 140) you seem to suggest this is possible. So, I am concluding yes, but would appreciate some more details as it is

critical to interpreting the work.

The references have been added.

Mineral dissolution is unlikely to affect ^{228}Ra flux because of its long half-life, i.e., long time is required to regenerate ^{228}Ra (unlike ^{224}Ra , see Garcia-Orellana et al., 2014 (doi.org/10.1016/j.gca.2014.05.009) and Moore et al., 1996 (doi.org/10.1016/0278-4343(95)00049-6)).

In line 140, we mean that the diffusion flux can vary depending on sediment type (i.e., Th concentrations and nature of the sediment) As per an earlier comment, high ^{228}Ra diffusion from shelf sediments in this region and/ or an addition source of Ra such as submarine groundwater discharge are the most likely sources. See answer to reviewer 1 and 3.

[Line 132] “ ***This suggests that either (i) the dissolved ^{228}Ra concentration in the Congo River is exceptionally high compared to other large rivers (~ 79 dpm 100 L^{-1} , vs <20 dpm 100 L^{-1} elsewhere); (ii) ^{228}Ra diffusion from shelf sediments in this region is anomalously high compared to other regions globally; or (iii) there is another source of Ra such as submarine groundwater discharge^{34, 35}. Based on observations elsewhere of large variability of ^{228}Ra diffusion from shelf sediments³⁶ and SGD input¹⁴, (ii) and (iii), or a combination of both, are most likely.*”**

6. Line 45: Please clarify this figure and mark very clearly, on the diagram itself perhaps, the 3 groups of samples that are discussed: the “Congo Shelf Zone”, the “Coastal Transect”, and the “Offshore Plume”. While I think all the info can be gathered from the text as it stands, it did take me a couple of reads to really nail these down.

Thank you for the suggestion. Figure 1 has been changed, and these points are now addressed. See also the legend of the figure.

7. Line 48: Add “offshore” before “3 degrees S”, and perhaps one line that says this was the seasonal high-flow period.

Line 48 was the legend of the figure 1 in the first version of the manuscript. Figure 1 and its legend has been changed. A line about the seasonal high-flow period has been added, as suggested.

[Line 145] ***“Sample collection occurred during the high discharge season of the Congo (...).”***

See also response to reviewer 1 about seasonal variability.

8. Line 67-78: I would move a version of the sentence “low removal primarily reflects...” to the beginning of this paragraph (e.g. before “Extrapolating” in line 57), because otherwise it appears as if you are trying to state as the primary point of this section that only 50% of the iron is removed. It sets the reader up for confusion.

The text has been changed for clarification.

[Line 47] ***“On the shelf where Congo waters first encounter the Atlantic Ocean, hereafter the “Congo-shelf-zone”, the mean dFe concentration was ~15% of the Congo River concentration, indicating low apparent dFe removal compared to River Congo freshwater. About 50-85% of river-derived dFe is reportedly removed from solution at low salinities (0-5) in the Congo estuary, with the greatest removal in large size fractions². Mean (\pm standard deviation) dFe concentration in the River Congo freshwater (averaged over 2017) was $7,380 \pm 3,150$ nM, similar to limited previous measurements ($\sim 9,000$ nM)². Extrapolating the linear regression line of dFe vs. salinity in the Congo-shelf-zone (Fig. 2a) to zero salinity provides an effective-zero-salinity-endmember concentration of $3,910 \pm 610$ nM ($R^2 = 0.76$), indicating that only ~50% of dFe is removed during estuarine mixing processes. This is consistent with prior work², but notably limited compared to other river systems where 90-99% is typically stripped from the water column^{2, 6}.”***

9. Line 59: Saying that you are extrapolating to a “zero salinity endmember” is also confusing, because this would generate a that technically occurs prior to any estuarine flocculation processes (that happen at salinities > 0). Instead, extrapolation of the data points in Figure 2A actually reveals the estuarine endmember that should have already undergone mixing and hosted flocculation processes. I would support language, “extrapolating to an estuarine endmember”

Indeed, we used this approach exactly to account for all the estuarine processes causing non-conservative behavior at salinity > 0 (as discussed in the classical application/derivation of the approach: Officer 1979, Boyle et al 1974, Liss 1976, Hydes and Liss 1977). The ‘estuarine’ endmember, for example, would be very different because of those processes. Therefore, we prefer to retain our original wording.

10. Line 74: Are these ratios specific to the Congo? Please state if so. Otherwise, I don’t think this is really a valid exercise given how variable river concentrations can be.

Those ratios were calculated from our Congo data. The text has been revised for clarification.

[Line 68] ***“Covariations of dissolved manganese (dMn) and cobalt (dCo) with salinity indicate an effective-zero-salinity-endmember higher than dMn and dCo concentrations measured in the river (Table 1). This is also indicative of non-conservative inputs in the Congo-shelf-zone relative to simple mixing of river and seawater (Supplementary Fig. 1). An additional TE source in the Congo-shelf-zone is also evident in the lower Fe:Mn (6.3 ± 6.0) and Fe:Co (525 ± 490) ratios compared to the Congo River (Fe:Mn = 71.2 ± 37.5 ; Fe:Co = 4.29 ± 2.34). These ratios reflect how dFe is removed relative to the other elements, which is unclear from fluxes alone. These ratios suggest that River Congo dFe is removed by a factor of 10, whereas the fluxes (Table 1, discussion below) indicate that the net removal is only a factor of 2. In summary, a multi-element approach also corroborates significant dTE inputs into the Congo-shelf-zone other than River Congo water.”***

11. Line 75: Add “relative to seawater background” before “in the Congo”?

The word “elevated” in the text refers to the surrounding water. We think that “background” is a little ambiguous and difficult to specifically define, so we prefer to keep the original wording.

12. Line 78 and others: I went to the cited reference because I was questioning the logic here, and I don’t think the cited paper supports the argument in this manuscript: “The Congo River plume may be classified as surface-advected (Yankovsky & Chapman, 1997); buoyant river inflow remains on top of shelf water forming a thin layer that, decoupled from bottom stress,

spreads many hundreds of kilometres offshore.” Now, just because turbulent benthic interactions are minimal offshelf, this doesn’t mean upwelling cannot be significant off-shelf. In fact, the cited paper proceeds to state in the next paragraph: “An estuarine type circulation has been identified here, where surface outflow is compensated at depth by a residual up-canyon transport of saline water (Denamiel, Budgell, & Toumi, 2013; Eisma & Van Bennekom, 1978).” I would remove or rephrase line 78 – I don’t think it supports the argument.

We argued that benthic input supplies Ra and TEs between the river mouth and the Congo-shelf-zone, but conservative Ra mixing behavior indicates no additional inputs beyond the Congo-shelf-zone. Nevertheless, the line has been removed as suggested.

Here is where I hope to ignite some thought: the Congo River is the only large river in the world still connected to its deep sea fan via a turbidite canyon, and because of these turbidity flows, the Congolobe expedition observed incredibly high rates of accumulation of organic/iron minerals (>2 cm/year) and high rates of iron solubilization from shelf-break down to deep-sea fan sediments (consistent with those in Fe-rich estuaries) up to 800 km offshore (Beckler et al. 2016; Taillefert et al. 2017; Rabouille et al. 2017 and other references therein). Under conditions of intense iron-diagenesis in sediments, a corresponding flux of Fe and TE typically also occurs, so I suspect that these processes, i.e. upwelled slope/deep-sea waters previously in contact with these sediments, may also be contributing to plume enrichments (in addition to shelf sediments as stated in line 140). The actual offshore transect in this manuscript processed to the North of the Congo River canyon. I wonder if during other times of year when the plume may be situated more southward, if we may expect even a higher contribution of sediment-derived Fe within the plume, and overall contributing to the gyre?

This process could indeed be important, but it is not the dominant feature that explains our observations, because this upwelling source would likely bring high Fe, but probably not high Ra, or Co, etc. The key point here is that we see elevated levels of Fe, Ra, and other trace elements (e.g. Mn and Co) and they all co-vary, indicating a common source. We appreciate the reviewer’s thoughtful input. Additional text has been added to the revised manuscript to reflect these points.

[Line 59] *“Alternatively, sources of Fe other than river-water may simply offset the loss from estuarine mixing. Indeed, similar ²²⁸Ra and other TE enrichments over the Congo-shelf-zone (Fig. 2b; Supplementary Fig. 1) suggest they have a common source, likely shelf-sediments^{3, 19, 20} or submarine groundwater discharge (SGD)⁵, indicating that the apparently low removal primarily reflects additional sources of dFe rather than unusually pronounced stabilization of the river-derived dFe. Strong benthic Fe input in this region is consistent with observed high rates of sediment accumulation and high rates of iron solubilization from the shelf-break down to deep-sea fan sediments (e.g., Beckler et al. 2016; Taillefert et al. 2017; Rabouille et al. 2017) which may also contribute to the relatively high dFe concentrations in the Congo-shelf-zone.”*

13. Some comments/alternative explanations relating to differences in Fe content, but similar salinities, for the CSZ and off-shelf sample sets (Figure 2):

a. Could the northern freshwater plumes be from a different river source and thus contain a different amount of Fe altogether?

The Congo is the dominant river in the region (approximately 4 times larger than the combined discharge of the small rivers to the north; Milliman and Farnsworth 2011), so we consider that the contribution of other rivers to a low salinity plume observed in the study region on the Western African shelf is minimal. We appreciate the comment and a text has been added to address this point.

[Line 196] *“Therefore, there is a strong indication that high Fe in this region is solely associated with the Congo outflow, as Fe, Mn and Co correlate strongly with Ra and salinity. In addition, as the Congo is the dominant river in the region (approximately 4 times larger than other more northern rivers combined³⁰), we consider that the contribution of other rivers to the low salinity plume observed in the study region is minimal.”*

b. Are the northern and southern sample plume water masses both from the Congo but from different time periods, as weekly variations in flow may result in various concentrations of iron in the river water?

Variation in the Fe endmember is possible/likely, but large short-term variation is unlikely due to coherence of the observed mixing patterns (e.g., Mn, Co, Ra) over very large spatial scales.

Also, we conclude that additional non-river sources contribute substantially to the high TE and Ra signals observed, so variation in the riverine Fe endmember would have limited effect on the estimated fluxes.

14. Line 99: I would remove this comparison, because the utility of the “zero salinity” end-member is virtually meaningless now that you have established that the dominant factor is indeed shelf input. This really had me confused.

We use this comparison to demonstrate the effect of estuarine processes leading to non-conservative mixing behavior. The effective zero salinity endmember (EZSE) calculation is necessary to make this comparison (see earlier response to reviewer 1 about the underlying theory).

15. Line 140: I suspect based on other thoughts throughout the paper that you believe here that explanation (ii) is the most logical. I do feel the atypical sedimentary environment of the Congo margin can explain this phenomenon.

We suspect that the most likely explanation is number (ii) or number (iii), or a combination of both. The unusual sedimentary environment is certainly interesting and potentially a contributing factor to high benthic inputs, but cannot explain the elevated concentrations of all dTEs measured- Please see earlier comment and response to the reviewer 1 above.

Reviewer #3

Review “Unprecedented Fe delivery by the River Congo to the South Atlantic Gyre” by L.H. Vieira

This study estimates fluxes of dissolved trace elements (Fe, Co, and Mn) based on ^{228}Ra data from the Congo shelf. The results of this study will be useful for the scientific community; however, I think that the paper needs to be significantly improved before considering it for publication. I am also wondering if Nature Communication Journal is the right place for this

study, because the method applied is not new, the authors presented a flux of trace elements from the Congo River that is new information but not original, and the short length of this Journal seemed to have prevented the authors to completely discuss the results. I have three main points that I think the authors should address:

The method may not be new, but it was not the focus of the paper. The results we present have major implications for the biogeochemical cycles in the Congo-influenced SE Atlantic Ocean. We think this is an important contribution to our understanding of the role of rivers in marine systems because it is fundamentally different from any other river system investigated globally. Although word count is limited, there is ample space within the limits to address the reviewer comments.

1) I noted some odd points in the method that I describe in details below. One point that needs to be addressed on the method is that the authors did not correct the ^{228}Ra average activity in the Congo-shelf-zone from offshore ^{228}Ra activities (if they did, it is not clearly written). This would affect the ^{228}Ra flux, which is then used for the TE fluxes. Therefore, the fluxes of the shelf would likely need to be re-calculated.

Good point. We redid the calculations using the offshore endmember (average concentration amongst stations in the offshore region – from station 1247 down along the meridian (0° E) (Vieira, 2019, dissertation), which had average salinity of 36.07 and $^{228}\text{Ra} = 1.78$ dpm/100 L. This implies that seawater can contribute ~12% of the 14.5 dpm/100 L observed at salinity ~30 (Fig 2a). This reduces the estimated flux by about 12%, but does not substantially change our original conclusions. In contrast to Ra, TE concentrations offshore are negligible compared with those in plume waters, and their fluxes are unchanged (Ra flux and TE/Ra ratio both change proportionally), although the uncertainties are now larger. It is important to mention that 10 samples from the cruise for Fe and Mn analysis were re-run to obtain a better standard deviation between repeated measurements compared to the initial values reported previously. The off-shelf flux slightly increased from 5.8×10^8 mol/yr to 6.8×10^8 mol/yr. Nonetheless, this does not materially change our flux calculations, or any of our conclusions. Yet as the updated values will be reported in texts elsewhere, we have updated the text and Source Data File accordingly for consistency.

2) More comparisons are needed regarding the fluxes of Ra228, Fe, Co, and Mn. An estimation of dust deposition should be included, especially after the recent study published by Mendez Barraqueta et al. 2019 based on data from the same cruise showing significant atmospheric deposition. The high dFe concentration on the Congo shelf might well be the results of the combination of high dust deposition, shelf sediment inputs, and large Congo River, rather than attributing the Fe in seawater solely to the Congo River inputs. Comparison of these fluxes with other study areas should also be included to answer the question: Is the Congo shelf a unique supplier of TE to the ocean?

A new paragraph has been added related to the dust deposition into our study region. As shown in the revised manuscript, atmospheric deposition contributes very little to the dFe concentration in our study region and, as noted, the flux calculated is likely an over-estimate as it assumes Aluminum in the region of interest arises exclusively from dust, whereas in reality a fraction likely also originates from the river. Also, we showed that Fe correlates with salinity, which indicates that dust could only have a minor effect on Fe concentration compared to river-associated sources. Furthermore, all elements have a common source, including Ra, which also indicates that this source is probably not dust. The new paragraph reads as:

[Line 176] ***“Atmospheric deposition is thought to be an important source of dFe to the surface ocean and the West African margin receives dust fluxes which are amongst the highest in the world. So how does atmospheric deposition compare to lateral dFe supply from the Congo across this region? Dissolved Al is a useful tracer of recent dust deposition to the surface ocean, and atmospheric deposition has been estimated from GA08 Al data⁴⁶ as $2.65 \text{ g m}^{-2} \text{ yr}^{-1}$ for the offshore (3° S) section and $2.67 \text{ g m}^{-2} \text{ yr}^{-1}$ for the coastal transect. The offshore values exclude stations within the coastal shelf zone where other dAl sources, certainly including a contribution from direct Congo discharge, preclude the use of this tracer. Nevertheless, if all dAl within the study region were attributed to atmospheric deposition it would correspond to deposition of $26.2 \text{ g m}^{-2} \text{ yr}^{-1}$ ⁴⁶. The fractional composition of dust and the elemental solubility within dust vary widely. Using a broad range of plausible dust Fe content (1.9-5.0%) and Fe solubility (0.14-21%)^{47, 48} suggests that Fe deposition in these regions is within the range of $13\text{-}4900 \text{ } \mu\text{mol m}^{-2} \text{ yr}^{-1}$ (a minimum and maximum limit given the contribution of dAl from non-atmospheric sources in this zone), or $0.01\text{-}4.93 \times 10^7 \text{ mol yr}^{-1}$, 1-3 orders of magnitude***

lower than our estimated dFe fluxes from the Congo-shelf-zone to the same region (Table 1).”
See references in the revised manuscript.

3) In my opinion, the title, abstract, and intro needs to be re-written a little. I personally don't like the catchy Nature forced title “unprecedented Fe delivery”. An alternative title could be in the lines of “the Congo shelf represents a unique (or major) supplier of TE to the South Atl...” I am confused by the title/Abstract/intro that focus on the Congo River.

We appreciate the suggestion; other reviewers supported the title with slight changes. It now reads: “Unprecedented Fe delivery from the River Congo margin to the South Atlantic Gyre”.

The TE fluxes are estimated from Ra228, which is a sediment tracer (except if the Congo river has exceptionally high 228Ra activity, which hasn't been demonstrated in this study). The fluxes are estimated using Ra for the “Congo-shelf-zone” and on an “off-shelf transect”, which include inputs from both the river and from sediment. The salinity range is pretty high, mainly above 32, which suggest less than 20% of freshwater contribution in the study zone.

This is true, and most of this is discussed in the manuscript. We argue that there must be source (s) of 228Ra between the Congo River mouth and the Congo-shelf-zone. As noted in our response to the other reviewers, we show that either (or both) high 228Ra diffusion from shelf sediments in this region and/ or an additional source of Ra such as submarine groundwater discharge are the most likely sources; the shelf-influenced-river waters are transported offshore. The fact that such high Ra, Fe and other dTE concentrations are found at such high salinities is a key point of the study as such concentrations are not observed so far offshore in other systems.

The author estimates the Congo river flux (discharge times concentration in the river), which is not necessarily comparable to the Ra flux because the latter is estimated for a specific area and not all the river freshwater discharge actually makes it into this restricted zone. I think there are a lot of discussion that has been cut probably because of the very short format of the journal, but I feel that it would improve the paper to provide more details regarding what these fluxes actually represent.

For any Ra work it is necessary to define an area of interest, and in the context of an oceanographic study this is the shelf region corresponding to our lateral transect, which was run in order to calculate offshore transport. The point of calculating both the Congo River flux and the Ra Congo-Shelf-Zone flux is to see if they are comparable and thus to what extent the River flux alone can explain concentrations on the shelf. Obviously, in addition to freshwater being laterally transferred west, the plume also disperses along the shelf generally in a northward direction. But the key finding of our study is that unusually large concentrations of dTEs are found at considerable distances off-shelf, and thus we focus on this lateral transport and not the fate of river water itself which isn't exported off-shelf.

[Line 292] ***“These two approaches were used to check if the Congo River flux and the ²²⁸Ra flux estimated for the Congo-shelf-zone were comparable, and thus see to which extent the River flux alone can explain the ²²⁸Ra concentrations on the shelf. Nonetheless, our study focuses on the transport of the Congo shelf-influenced-river waters enriched in ²²⁸Ra and TEs to offshore regions.”***

Methods:

- The second count for Ra224 was performed 6 weeks later whereas the second count for Th228 is usually done 2 weeks after collection. Was it really done after 6 weeks? Then some Bateman correction should probably be applied.

The fibers were first counted on-board the ship to determine the total activities for the short-lived Ra isotopes. The total activity of a daughter (e.g., ²²⁴Ra) consists of a component which is supported by secular equilibrium with its parent (i.e. ²²⁴Ra activity is equal the activity of its parent ²²⁸Th) and an “excess” component which decays with time. The Mn-fibers were aged for 6 weeks in order to allow excess ²²⁴Ra to completely decay (2 weeks are not enough for ²²⁴Ra to reach the equilibrium). The samples were then recounted to determine ²²⁸Th activities and thus correct the total ²²⁴Ra for supported activity:

$$^{224}\text{Ra}_{\text{ex}} = ^{224}\text{Ra}_{\text{total}} - ^{224}\text{Ra}_{\text{supported}}$$

No ingrowth or decay correction (i.e., using the Bateman equation) was necessary for the excess Ra calculation, although sample activities were corrected for any time delay between sample collection and measurement. The text has been revised to address this point:

[Line 235] ***“The fibers were counted onboard and aged for six weeks, in order to allow excess ²²⁴Ra to completely decay. They were then recounted to determine ²²⁸Th concentrations and thus correct the total ²²⁴Ra for the supported activity.”***

- After measurements of short-lived on the RaDeCc, the authors said that the fibers were ashed and then leached. Did the author leached the ashes? I am aware of two methods: ash (Charette et al. 2001) or leach/co-precipitate (Moore method) the fibers but I have never seen both processes applied one after the one. Is it a typo by the authors or was it the real procedure?

This was indeed the actual procedure. Additional text has been added to clarify the methodology:

[Line 239] ***“After measurement of ²²⁴Ra, fibers were ashed and subsequently leached in order to determine the activity of long-lived Ra (²²⁸Ra and ²²⁶Ra) isotopes using a high-purity, well-type germanium (HPGe) gamma spectrometer. As the remaining amount of ash we obtained was too large to fit inside the well of the HPGe detector (Canberra Eurisys GMBH, EGPC 150), the ashes were subsequently leached followed by co-precipitation with BaSO₄. Ashing the fibers before leaching produced a more homogeneous material that was easier to handle. Although atypical, this method has been used elsewhere⁶⁴. The fibers were ashed at 600° C for 20 h, leached in 6 M HCl followed by co-precipitation with BaSO₄.”***

- The Congo-self-zone flux of ²²⁸Ra was estimated as inventory/residence time. The authors used the excess ²²⁸Ra in the flux calculation. The excess Ra²²⁸ should be the average ²²⁸Ra concentration in the study area corrected from the offshore ²²⁸Ra concentration. This is to account for mixing with offshore waters that have a background ²²⁸Ra. If the authors had used ²²⁴Ra, there would be no need to consider the excess ²²⁴Ra because offshore waters have negligible amount of ²²⁴Ra. The authors should use an excess ²²⁸Ra of 14.5 – 4 = 10.5 based on figure 2, the ²²⁸Ra is about 4 dpm/100L outside of the congo shelf zone. Or 8.6 dpm/100L considering the average offshore activities from the offshelf transect on table 1, it depends on what this offshore average value is. This will change the fluxes of TE.... Cf Moore, W. S.: Inappropriate attempts to use distributions of ²²⁸Ra and ²²⁶Ra in coastal waters to model

mixing and advection rates, *Continental Shelf Research*, 105, 95–100, doi:10.1016/j.csr.2015.05.014, 2015.

As per earlier comments, we corrected the ^{228}Ra average activity in the Congo-shelf-zone.

Other comments:

- What is the input of atmospheric deposition of Fe to the study area? What is the contribution of dust to the high Fe concentration in the Congo River plume? Dust is a significant source of Fe to the ocean (Jickells et al., 2005), therefore, the contribution of dust-derived Fe in the surface concentration cannot be ignored. Especially considering a recent publication from Menzel Barraqueta et al. (2019) showing that the study area receives one of the highest atmospheric deposition rates to the Atlantic Ocean calculated from Al data from the same cruise GA08 (~8-10 g/m²/y). A rough estimate of the flux of Fe from dust can be calculated using the average atmospheric deposition, the concentration of Fe if measured on aerosol during the cruise or the composition of Fe in crust, and the solubility factor. Similar calculations can be done for Mn, and Co shown in the suppl. info. A more precise location can also be done since the deposition rate is estimated for the same stations. The author could thus perform a mass balance and potentially correct the Fe concentration that is due to dust.

The atmospheric deposition estimate suggested by the reviewer is now included (see our earlier response). However, it is worth noting that high mass fluxes for GA08 (GUIN) are very high because of river input as Menzel-Barraqueta et al., 2019 argued (the dAl flux from the river is 10 times larger than from dust). In addition, there are two very clear reasons for concluding that dust was a minor dTE source compared to river-associated sources across this region. First, TEs correlate with salinity in our study, which strongly suggests they have a riverine source. Second, atmospheric deposition does not contribute to the Ra signal in the oceans and, because all the elements seem to have the same origin, this indicates that the main TE source is not dust. The extent to which atmospheric sources (rain/dust) influence the freshwater concentration of dTEs in the River Congo is beyond the scope of this study.*

- The comparison of the Fe:Mn ratio in the river to the Fe:Mn ratio in the shelf-zone sounds a bit off to me, but I might be wrong. The lower Fe:Mn ratio in the shelf could simply reflect a different behavior of Fe and Mn. Fe is removed faster than Mn, thus the faster removal of Fe

would result in a lower Fe:Mn ratio in the shelf compare to the Fe:Mn ratio in river without involving additional source as suggested by the authors.

An additional source is potentially indicated by the difference in ratios, although as noted this strongly depends on how Fe is removed relative to the other elements. The ratios suggest that dFe would have to drop by a factor of 10 to produce the observed shift, whereas the fluxes indicate that the net removal is only a factor of 2. Therefore, it doesn't seem that the ratios only reflect estuarine mixing or net removal. An additional line has been added to clarify this point:

[Line 71] ***“An additional TE source in the Congo-shelf-zone is also evident in the lower Fe:Mn (6.3 ± 6.0) and Fe:Co (525 ± 490) ratios compared to the Congo River (Fe:Mn = 71.2 ± 37.5 ; Fe:Co = 4.29 ± 2.34). These ratios reflect how dFe is removed relative to the other elements, which is unclear from fluxes alone. These ratios suggest that River Congo dFe is removed by a factor of 10, whereas the fluxes (Table 1, discussion below) indicate that the net removal is only a factor of 2. In summary, a multi-element approach also corroborates significant dTE inputs into the Congo-shelf-zone other than River Congo water.”***

- Can the authors discuss the three hypothesis that could explain the high Ra flux from the Congo-shelf-zone. 1) exceptionally high Ra in the Congo, 2) high Ra diffusion from sediment, 3) SGD. Which one is the most likely? Why Ra would be much higher in the Congo River compare to other riverine system.

This point is now clarified in the text as also suggested by reviewer 1 and 2 (see above). As mentioned previously, the most likely explanation are numbers (ii) and (iii), or a combination of both, because ^{228}Ra diffusion from shelf sediments has large regional variability and the existence of another source of Ra such as submarine groundwater discharge in a large river system such as the Congo is plausible. We can't exclude number (i), but agree with the reviewer that it is unlikely because global rivers do not show large variation in Ra activity, and the Ra concentration in the Congo would have to be higher by factor of 4 to support the observed flux.

- How does the Ra and TE fluxes from the Congo shelf zone compared to fluxes estimated in other regions? This could be compared to the TE fluxes to the north Atlantic Ocean from Charette, M. A., Lam, P. J., Lohan, M. C., Kwon, E. Y., Hatje, V., Jeandel, C., Shiller, A. M.,

Cutter, G. A., Thomas, A., Boyd, P. W., Homoky, W. B., Milne, A., Thomas, H., Andersson, P. S., Porcelli, D., Tanaka, T., Geibert, W., Dehairs, F. and Garcia-Orellana, J.: Coastal ocean and shelf-sea biogeochemical cycling of trace elements and isotopes: lessons learned from GEOTRACES, *Phil. Trans. R. Soc. A*, 374(2081), 20160076, doi:10.1098/rsta.2016.0076, 2016. *Our dFe flux, for example, from the Congo shelf zone is an order of magnitude higher than reported elsewhere (e.g., Sanial et al., 2018; Charette et al., 2016). The total off-shelf Fe flux, is consistent with Sanial et al., 2018; Charette et al., 2016, because the area over which the flux occurs is 1-2 orders of magnitude smaller. This point has been added to the revised manuscript:*

[Line 138] ***“Radium-228 and TEs have a common source in the estuarine mixing zone up to our Congo-shelf-endmember, and combining the ^{228}Ra flux and concentration ratios of TE: ^{228}Ra (Methods) for the Congo-shelf-zone provides a dFe flux ($d\text{Fe-Flux}_{\text{Congo-shelf}}$) of $5.6 \pm 4.6 \times 10^5 \mu\text{mol m}^{-2} \text{yr}^{-1}$ ($5.6 \pm 4.6 \times 10^9 \text{mol yr}^{-1}$), an order of magnitude higher than reported along other continental margins^{37, 38}. The corresponding $d\text{Mn-Flux}_{\text{Congo-shelf}}$ was $4.4 \pm 1.8 \times 10^8 \text{mol yr}^{-1}$; and $d\text{Co-Flux}_{\text{Congo-shelf}}$ was $5.3 \pm 2.0 \times 10^6 \text{mol yr}^{-1}$ (Table 1).”***

[Line 164] ***“Similarity between the TE and ^{228}Ra distributions suggests that the fluxes scale proportionally³⁸. Given a Fe: ^{228}Ra ratio of $0.1 \text{pmol atom}^{-1}$, the off-shelf dFe-Flux from the Congo River plume into the South Atlantic Ocean is $6.8 \pm 2.3 \times 10^8 \text{mol yr}^{-1}$ (or $138 \pm 51 \text{mol m}^{-2} \text{yr}^{-1}$). On a global scale, this represents 0.7-2.3% of the global sedimentary Fe flux ($2.7 - 8.9 \times 10^{10} \text{mol yr}^{-1}$)^{3, 41}, or approximately $40 \pm 15\%$ (based on the uncertainties of our estimate) of total dFe atmospheric deposition into the entire South Atlantic Ocean⁴². The similarity between dMn and dCo fluxes in the Congo-shelf-zone and off-shelf transect (Table 1) is consistent with their conservative behavior along the Congo River plume, likely because of slow Mn and Co oxidation^{43, 44} and Mn photo-reduction in surface waters⁴⁵ which keeps Mn in solution and facilitates its transport to the open ocean. Our total off-shelf TE fluxes are similar to those reported elsewhere for other ocean margin regions^{37, 38}, although in the current study this flux occurs over a much smaller area .”***

- Another approach to estimate the Congo river input could be used. The authors used the TE concentrations measured in the river and the river discharge. An estimation of the amount of

freshwater actually entering the Congo-shelf zone can be done using the average salinity and applying conservative mixing of salinity to calculate the volume of freshwater instead of using the river discharge. Doing so, the source of TE from the freshwater Congo river will be directly comparable to the sediment derived estimate of Ra (since the mol/y unit implies a specific surface considered).

The method suggested by the reviewer would likely work in a more completely sampled river plume. Our limited sampling coverage in the Congo-shelf-zone (due to restrictions by Angola and local oil/gas platform operators) would lead to unacceptably high uncertainty.

Unfortunately, satellite-derived surface salinity data shown in the first version of this manuscript doesn't match well with the measured salinity over the cruise track (Fig. 1) as mentioned in the manuscript. Uncertainties in the plume thickness (e.g., Fig S4) would have to be considered as well.

- The authors ruled out the benthic supplies of Ra and TE because of the shallow Congo river plume. But we don't really know what is the water column depth along the transect. Since the plume is moving along shore, it could be hugging the coast in shallow water. Thus, it would be useful to know the water column depth compare to the 15 m of river plume. Similarly, the odv plot figure 4 in suppl. info has no bathymetry, how steep is the shelf? Maybe adding the bathymetry to the Figure 1 would already give some information.

Thank you for the suggestion. Figure 1 has been changed and the bathymetry is now added. The previous figure 1 has been moved to the supplementary material.

- Can the author discuss the difference of Fe flux between the Congo shelf zone and the off shelf transect? Does this provide information on the scavenging rate or biological uptake of Fe in between the two zones?

The biogeochemical processes controlling the distribution of dFe in our study region are still unclear (but will include scavenging and biological uptake). This point will hopefully be clarified in future publications about the GA08 cruise when full Fe speciation data is available and biological activity and particle fluxes for the region are better constrained

- Can the author discuss the exceptionally high dFe concentration 600 km from the Congo river mouth? Is the Fe 600 km away from the mouth river from the river or does it has another origin? That is just a side comment that I am curious about.

There is a strong indication that Fe is associated with the Congo outflow, as dFe correlates strongly with both Ra and with salinity. As discussed above, other sources (e.g., dust) appear to be more than two orders of magnitude too low to account for the source. We cannot think of any other hypothesis which would be consistent with observed distributions of both Ra and Fe over the transect. We added a text about that.

Comment on figures:

- Figure 1: I would add a large scale map of Africa for example in a subplot to better situate the study area, to be more visual.

Could the authors show the salinity underway data to better visualize the actual plume during the survey? Should be better than satellite image, especially because the authors acknowledge a difference in salinity between satellite and in situ data. Add the bathymetry to see the width and steepness of the shelf.

Thank you for the suggestion. Figure 1 has been changed as suggested

- In general, I think that surface plots would be useful and could be added as subplots to the figure 2 for example since the cruise track is composed by an along shore transect and an offshore transect.

This will be useful, but as we need to show the mixing line on scatter plots for quantitative discussion, we prefer not to add qualitative plots. The sharp changes in dTE concentrations due to freshwater 'pockets' offshore also make interpolated surface plots undesirable due to the artificial gradients created between such data points.

- Figure 2: switch panels c and d to show Fe in a) and c), and Ra in b) and d)

We numbered the graphs according to the order they are mentioned in the text. So, to follow the text organization, we prefer to keep the current format.

- Figure 3: the Congo data do not stick out, I would exchange the symbol between Congo and Mississippi to highlight the data from this study in red. Based on this graph, the ^{228}Ra are consistent with other estuaries. Could this help in the discussion of the 3 hypotheses about the high flux of ^{228}Ra I mentioned earlier?

The reviewer is referring to figure 3 of the supplementary info. The figure has been changed as suggested. This is true that the ^{228}Ra are consistent with other estuaries in high salinity (and this is why we don't believe in hypothesis number (i) (see discussion above and text below)). Nonetheless, without directly sampling the river endmember and lower salinities, it is not possible to conclusively rule out the possibility of an unusually high river Ra endmember concentration.

[Line 132] ***This suggests that either (i) the dissolved ^{228}Ra concentration in the Congo River is exceptionally high compared to other large rivers (~ 79 dpm 100 L^{-1} , vs <20 dpm 100 L^{-1} elsewhere); (ii) ^{228}Ra diffusion from shelf sediments in this region is anomalously high compared to other regions globally; or (iii) there is another source of Ra such as submarine groundwater discharge^{34, 35}. Based on observations elsewhere of large variability of ^{228}Ra diffusion from shelf sediments³⁶ and SGD input¹⁴, (ii) and (iii), or a combination of both, are most likely.***

REVIEWERS' COMMENTS:

Reviewer #1 (Remarks to the Author):

I am satisfied with the changes that have been made to the paper.

There are some minor grammatical errors so the text should be carefully checked by a native English speaking scientist.

Line 48 and 73: River Congo and Congo River are used close to each other in the text. Is that intentional, or should there be some consistency?

Reviewer #2 (Remarks to the Author):

Thank you for considering my comments, all which have been sufficiently addressed. The manuscript is much more concise now and is an important contribution to our field.

Jordon Beckler

We would like to once more thank the reviewers for the constructive comments and suggestions they made to improve this article. Our responses to the reviewers' comments are written in italic.

Reviewers' comments:

Reviewer #1

I am satisfied with the changes that have been made to the paper.

There are some minor grammatical errors so the text should be carefully checked by a native English speaking scientist.

Line 48 and 73: River Congo and Congo River are used close to each other in the text. Is that intentional, or should there be some consistency?

The text has been revised throughout and the wording corrected to maintain consistency.

Reviewer #2

Thank you for considering my comments, all which have been sufficiently addressed. The manuscript is much more concise now and is an important contribution to our field.

Thank you. We appreciate the reviewer comment.